# Quorum sensing controls *Vibrio cholerae* multicellular aggregate formation

Matthew Jemielita[1], Ned S Wingreen[1], Bonnie L Bassler[1,2]*

[1]Department of Molecular Biology, Princeton University, Princeton, United States; [2]Howard Hughes Medical Institute, Chevy Chase, United States

**Abstract** Bacteria communicate and collectively regulate gene expression using a process called quorum sensing (QS). QS relies on group-wide responses to signal molecules called autoinducers. Here, we show that QS activates a new program of multicellularity in *Vibrio cholerae*. This program, which we term aggregation, is distinct from the canonical surface-biofilm formation program, which QS represses. Aggregation is induced by autoinducers, occurs rapidly in cell suspensions, and does not require cell division, features strikingly dissimilar from those characteristic of *V. cholerae* biofilm formation. Extracellular DNA limits aggregate size, but is not sufficient to drive aggregation. A mutagenesis screen identifies genes required for aggregate formation, revealing proteins involved in *V. cholerae* intestinal colonization, stress response, and a protein that distinguishes the current *V. cholerae* pandemic strain from earlier pandemic strains. We suggest that QS-controlled aggregate formation is important for *V. cholerae* to successfully transit between the marine niche and the human host.
DOI: https://doi.org/10.7554/eLife.42057.001

*For correspondence:
bbassler@princeton.edu

Competing interests: The authors declare that no competing interests exist.

## Introduction

Quorum sensing (QS) is a cell–cell communication process that bacteria use to orchestrate collective behaviors. QS relies on the production, release, and group-level detection of molecules called auto-inducers (reviewed in *Papenfort and Bassler, 2016*). At low cell density (LCD), when autoinducer concentration is low, QS promotes gene expression programs that benefit individual bacteria. At high cell density (HCD), when autoinducer concentration exceeds the threshold required for detection, QS drives gene expression programs beneficial to the community.

*Vibrio cholerae* is the etiological agent of the disease cholera. In *V. cholerae*, QS controls virulence factor production and biofilm formation (*Hammer and Bassler, 2003*; *Miller et al., 2002*; *Papenfort and Bassler, 2016*; *Zhu and Mekalanos, 2003*; *Zhu et al., 2002*) (*Figure 1*). *V. cholerae* relies on two major autoinducers: CAI-1, an intra-genus-specific autoinducer (*Higgins et al., 2007*; *Kelly et al., 2009*; *Miller et al., 2002*), and AI-2, an autoinducer broadly conserved across bacteria and used for inter-species communication (*Chen et al., 2002*; *Schauder et al., 2001*). The CAI-1 and AI-2 receptors are CqsS and LuxPQ, respectively (*Bassler et al., 1994*; *Miller et al., 2002*; *Neiditch et al., 2005*). In the absence of autoinducer, CqsS and LuxPQ act as kinases funneling phosphate through an integrator protein, LuxU, to LuxO, the shared response regulator protein (*Freeman and Bassler, 1999*). Phosphorylated LuxO activates transcription of genes encoding four small RNAs: Qrr-1 to Qrr-4 (*Lenz et al., 2004*). Qrr1-4 activate translation of AphA and repress translation of HapR; respectively the master LCD and master HCD QS regulators (*Lenz et al., 2004*; *Rutherford et al., 2011*). Thus, at LCD, AphA is made and HapR is not, and *V. cholerae* cells act as individuals (*Figure 1A*). When bound to their cognate autoinducers, which occurs at HCD, CqsS and LuxPQ switch from acting as kinases to acting as phosphatases, dephosphorylating LuxO, via LuxU. Dephosphorylated LuxO is inactive, so transcription of *qrr*1-4 does not occur. In the absence of the Qrr sRNAs, activation of AphA translation ceases and HapR translation is no longer repressed. Thus,

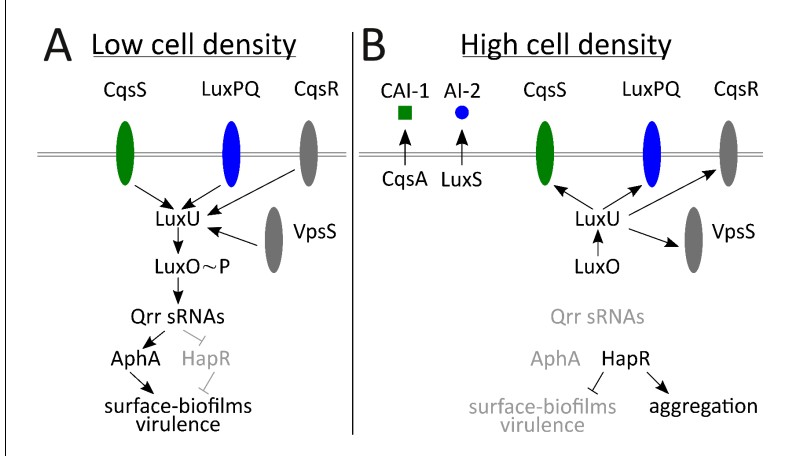

**Figure 1.** Simplified *V. cholerae* quorum-sensing circuit. (**A**) At low cell density (LCD), when autoinducer concentration is low, the transmembrane receptors CqsS (green) and LuxPQ (blue) act as kinases and funnel phosphate to LuxO through the intermediary protein LuxU. Phospho-LuxO activates transcription of genes encoding regulatory RNAs called the Qrr sRNAs. The Qrr sRNAs activate translation of AphA and repress translation of HapR. This condition promotes the LCD QS program, which includes expression of genes encoding virulence factors and surface-biofilm formation. (**B**) At high cell density (HCD), when autoinducers have accumulated, CAI-1 (green squares) and AI-2 (blue circles) bind to their respective cognate receptors, CqsS and LuxPQ. Autoinducer binding converts the receptors into phosphatases that dephosphorylate and inactivate LuxO. Therefore, the Qrr sRNAs are not produced. In the absence of the Qrr sRNAs, AphA translation is not activated and HapR translation is not repressed. HapR represses the surface-biofilm and virulence programs. HapR activates the aggregation process that occurs in liquid. Two other QS receptors, VpsS and CqsR (both depicted in gray), with unknown ligands, also transduce QS information through LuxU.

DOI: https://doi.org/10.7554/eLife.42057.002

HapR is made and AphA is not, and *V. cholerae* cells engage in group behaviors (*Figure 1B*). Two other QS receptors, CqsR and VpsS, with unknown ligands, also convey information into the QS circuit via LuxU (*Jung et al., 2015*; *Shikuma et al., 2009*).

In *V. cholerae*, QS controls the formation of surface-bound multicellular communities called biofilms (*Hammer and Bassler, 2003*; *Teschler et al., 2015*; *Zhu and Mekalanos, 2003*). Specifically, AphA promotes and HapR represses the production of four components required for biofilm formation: vibrio polysaccharide (VPS) and three matrix proteins, RbmA, Bap1, and RbmC (*Hammer and Bassler, 2003*; *Yang et al., 2010*; *Yildiz et al., 2001*). The result of this regulatory arrangement is that *V. cholerae* forms biofilms at LCD and disperses from them at HCD (*Miller et al., 2002*; *Singh et al., 2017*; *Teschler et al., 2015*). Extracellular DNA (eDNA) also contributes to *V. cholerae* biofilm formation (*Seper et al., 2011*). eDNA levels are regulated by the activity of two extracellular nucleases: Xds and Dns, the latter of which is repressed at HCD by HapR (*Blokesch and Schoolnik, 2008*).

The lifecycle of *V. cholerae* requires transitions between the marine environment and the human host (*Almagro-Moreno et al., 2015*). In both niches, biofilm formation and dispersal appear to occur. Specifically, in the marine environment, *V. cholerae* associates with chitin, a major component of transient nutrient sources such as marine snow (*Huq et al., 1983*; *Pruzzo et al., 2008*;

*Yawata et al., 2014*). To successfully adapt to the disappearance of marine snow upon consumption, *V. cholerae* must be able to transition between the surface-associated state and the planktonic state. Infections in humans occur when contaminated water or food containing planktonic and/or aggregates of *V. cholerae* cells is ingested. Early in human infection, *V. cholerae* cells, after passaging through the stomach, transit from the lumen of the small intestine through the mucosal layer to the epithelial surface (*Almagro-Moreno et al., 2015*). Both motility and chemotaxis are reported to play roles in this process suggesting that cells exist in the planktonic state as they make this transition (*Butler and Camilli, 2005*; *Liu et al., 2008*). At the epithelial surface, *V. cholerae* represses motility, forms surface microcolonies, and activates its virulence program (*Almagro-Moreno et al., 2015*; *Matson et al., 2007*; *Millet et al., 2014*; *Watnick et al., 1999*). Virulence factors, including the toxin-coregulated pilus (TCP) and cholera toxin (CT), are produced at LCD. CT causes severe diarrhea characteristic of the disease cholera (*Matson et al., 2007*). Later in infection, at HCD, HapR represses virulence factor production (*Miller et al., 2002*; *Zhu et al., 2002*), and HapR, together with RpoS, launches a mucosal escape program (*Nielsen et al., 2006*). Specifically in the case of QS, HapR activates expression of *hapA*, encoding the HapA mucinase, reported to contribute to the host-escape process (*Booth et al., 1983*; *Silva et al., 2003*; *Zhu et al., 2002*). *V. cholerae* is shed back into the environment either as planktonic cells or as multicellular aggregates (*Faruque et al., 2006*; *Nelson et al., 2007*).

Elegant work has defined the mechanisms underlying *V. cholerae* biofilm formation on surfaces (*Absalon et al., 2011*; *Berk et al., 2012*; *Fong et al., 2017*; *Fong et al., 2010*; *Watnick et al., 1999*; *Yan et al., 2016*), however, relatively little attention has been paid to whether *V. cholerae* forms communities in the absence of surfaces. To our knowledge, two exceptions are studies of pilus-mediated autoagglutination by either TCP in the classical biotype of *V. cholerae* or by the

DNA-uptake pilus (or, chitin-regulated pilus, ChiRP) in strain A1552 of the El Tor *V. cholerae* biotype (*Adams et al., 2018*; *Kirn et al., 2000*; *Taylor et al., 1987*). Given that the known lifecycle of *V. cholerae* includes stages in which the bacteria are not surface-associated, we explored whether *V. cholerae* forms communities in the absence of a surface with a focus on the role of QS in the process. We find that *V. cholerae* indeed forms communities in liquid. We call these non-surface-associated *V. cholerae* communities 'aggregates' to distinguish them from the familiar *V. cholerae* biofilms which grow on surfaces.

Mutant analyses demonstrate that components required for *V. cholerae* surface-associated biofilm formation are not required for aggregate formation. Moreover, aggregates form in the HCD QS-state, require the HCD master regulator HapR, and can be driven by the addition of exogenous QS autoinducers. By contrast, surface-associated biofilm formation occurs in the LCD QS-state, is repressed by HapR, and the HCD QS-state is anti-correlated with surface-biofilm formation (*Hammer and Bassler, 2003*; *Singh et al., 2017*). We show that aggregation occurs independently of cell growth, again distinguishing this process from surface-associated biofilm formation. We demonstrate that eDNA plays a structural role in aggregate formation, but that eDNA alone is not sufficient to induce aggregation. A genetic screen to identify components required for *V. cholerae* aggregate formation revealed genes including ones involved in stress response to nutrient limitation, phosphate acquisition from eDNA, and, notably, a gene unique to the current pandemic strain of *V. cholerae*. Combined, these results suggest that aggregation may be a strategy employed by *V. cholerae* to survive under starvation conditions, and this program may also contribute to the pandemic potential of the current *V. cholerae* biotype. Investigating *V. cholerae* aggregation may provide insight into how the bacterium successfully transitions between its different niches as well as reveal general mechanisms underlying non-surface-dependent multicellular community formation.

## Results

### *V. cholerae* forms multicellular aggregates in the HCD QS-state

As part of its natural lifecycle, *V. cholerae* must repeatedly transition between surfaces and the liquid phase. We wondered if, analogous to what occurs on surfaces, *V. cholerae* forms multicellular communities in liquid and, if so, what role QS plays, if any. To explore these possibilities, cultures of wild-type (WT), LCD QS-locked (LCD-locked), and HCD QS-locked (HCD-locked) *V. cholerae* strains were grown overnight under conditions of gentle shear in lysogeny broth (LB) supplemented with 10 mM of $Ca^{2+}$ to approximate the oceanic calcium concentration (*Kierek and Watnick, 2003a*). We generated strains 'locked' in the LCD and HCD QS modes by introducing single amino acid substitutions in LuxO at residue 61, the site of phosphorylation (*Freeman and Bassler, 1999*; *Hurley and Bassler, 2017*). LuxO D61E is a phosphomimetic and LuxO D61A cannot be phosphorylated. Thus, *V. cholerae* cells carrying *luxO D61E* and *luxO D61A* are locked in the LCD and HCD QS-states, respectively. For imaging purposes, we introduced constitutively expressed fluorescent reporter constructs onto the chromosomes of these strains and the strains described below. The fluorescent proteins are: mKO, mKate2, and mTFP1.

The LCD-locked strain formed clusters in liquid at 22 h (*Figure 2A*). We anticipated this result because LCD-locked *V. cholerae* cells are in a constitutive biofilm-forming state (*Hammer and Bassler, 2003*). However, to our surprise, the HCD-locked strain also formed multicellular communities in liquid. Moreover, these aggregates were larger and morphologically distinct from the LCD-locked clusters (*Figure 2B*). The WT strain produced aggregates morphologically similar to those formed by the HCD-locked strain (*Figure 2C*). Our finding that multicellular communities are produced by WT *V. cholerae* at HCD and by the HCD-locked QS strain is unexpected because on surfaces, HCD-locked *V. cholerae* cells do not form biofilms (*Hammer and Bassler, 2003*). For the remainder of this text, we refer to the communities that form in the LCD QS-state in liquid as clusters, the large multicellular communities that form in the HCD QS-state in liquid as aggregates, and communities that form on surfaces as surface-biofilms. The terms aggregates and biofilms are both used, sometimes interchangeably, in the literature. Other terms are used to describe more specific examples of bacterial aggregation (e.g., flocculation, co-aggregation, auto-aggregation, and autoagglutination) (*Bieber et al., 1998*; *Chiang et al., 1995*; *Flemming and Wingender, 2010*; *Rickard et al., 2003*).

We emphasize that we are using the term 'aggregates' simply to distinguish this program from the *V. cholerae* surface-biofilm program.

## HCD-QS aggregates are VpsL-independent

Essential for formation of *V. cholerae* surface-biofilms is VPS, a component of the extracellular matrix (*Fong et al., 2010*; *Teschler et al., 2015*). We assessed whether VPS is also required for the formation of aggregates by deleting *vpsL*, a gene required for VPS production (*Fong et al., 2010*). The *ΔvpsL* LCD-locked *V. cholerae* strain failed to form clusters (*Figure 2D*), demonstrating that VPS, in addition to driving the formation of surface-biofilms, also contributes to cluster formation in liquid. However, the *ΔvpsL* HCD-locked strain and the *ΔvpsL* WT strain formed aggregates similar to those made by the two parent strains possessing *vpsL* (*Figure 2D*). Thus, VPS is dispensable for aggregate formation in liquid. In the remainder of the experiments reported here, unless explicitly stated otherwise, all strains harbor the *ΔvpsL* mutation in order to distinguish aggregate formation from the VpsL-dependent surface-biofilm program. Aggregate formation does not require initial growth on a surface nor is it a consequence of our sampling protocol (*Figure 2—figure supplement 1*), and aggregates are finite sized (*Figure 2—figure supplement 2*, *Video 1*).

VPS-independent biofilms have been reported previously in *V. cholerae* O139 strain M010. Specifically, in medium supplemented with oceanic $Ca^{2+}$ levels, genes related to O-antigen synthesis (*wbfF* and *wbfR*) were found to be involved in development of VPS-independent surface-biofilms (*Kierek and Watnick, 2003b*; *Kierek and Watnick, 2003a*). The strain we use here, *V. cholerae* El Tor O1 C6706str2, belongs to a different serogroup than *V. cholerae* O139 strain M010 (*Blokesch and Schoolnik, 2007*; *Thelin and Taylor, 1996*). The growth conditions that we use for aggregate formation include oceanic $Ca^{2+}$ levels, so we examined whether *wbfF* and *wbfR* played any role in the process we are studying. We deleted the homologs of *wbfF* (*vpsN*, *vc0936*, 29% amino acid similarity) and *wbfR* (*asnB*, *vc0991*, 26% amino acid similarity). *Figure 2—figure supplement 3* shows that both the *ΔvpsL ΔvpsN* HCD-locked and the *ΔvpsL ΔasnB* HCD-locked mutants form aggregates as effectively as the *ΔvpsL* HCD-locked strain. Thus, *vpsN* and *asnB* do not contribute to the aggregation process we report here. We also note that *V. cholerae* O139 strain M010 is a natural variant that is locked in a LCD QS-state because of a mutation in *hapR* (*Joelsson et al., 2006*). Again, this feature is consistent with our current finding that it is the HCD-locked strain, not the LCD-locked strain, that forms the aggregates we are investigating. We also considered the possibility that $Ca^{2+}$ ions could play a signaling role in aggregate formation. To investigate this notion, we deleted *carR*, which encodes the $Ca^{2+}$ responsive two-component response regulator, CarR (*Bilecen and Yildiz, 2009*). Removal of *carR* had no effect on aggregate formation in the *ΔvpsL* HCD-locked strain (*Figure 2—figure supplement 3*). Lastly, the type IV TCP and the chitin-regulated ChiRP, respectively, contribute to autoagglutination in liquid in classical *V. cholerae* strains and to autoaggregation in El Tor *V. cholerae* strain A1552 (*Adams et al., 2018*; *Kirn et al., 2000*). There is one additional known type IV pilus in *V. cholerae*, the mannose-sensitive haemagglutinin (MSHA) pilus (*Watnick et al., 1999*), although no role associated with autoagglutination has been reported. Nonetheless, to be thorough, we deleted the major pilin subunits for all three pili. Specifically, we deleted *tcpA*, *pilA*, and *mshA* for TCP, ChiRP, and MSHA, respectively, all in the *ΔvpsL* HCD-locked strain. No loss of aggregation occurred in any case (*Figure 2—figure supplement 4*). Thus, the aggregate formation process that we are studying here does not require these major components identified previously to control formation of multicellular communities of *V. cholerae* either on surfaces or in liquid.

## Aggregation dynamics are rapid

We explored the kinetics of aggregate formation to compare the process to cell-division-driven surface-biofilm formation (*Yan et al., 2016*). Starting 16 h after inoculation, we sampled and imaged liquid suspensions containing the *V. cholerae* strains used in *Figure 2D*. We selected the 16 h time point to begin the analysis because prior to that time point all strains under study existed largely as planktonic cells. We sampled at 3 h intervals for 9 additional hours. The *ΔvpsL* LCD-locked strain showed no aggregation for the entire 25 h of analysis (*Figure 2E–H,Q*). At 16 h, both the *ΔvpsL* WT and the *ΔvpsL* HCD-locked strains consisted of planktonic single cells and some small clusters. However, there were no aggregates (*Figure 2I,M,Q,R*). Aggregation occurred by 19 h in the *ΔvpsL* WT

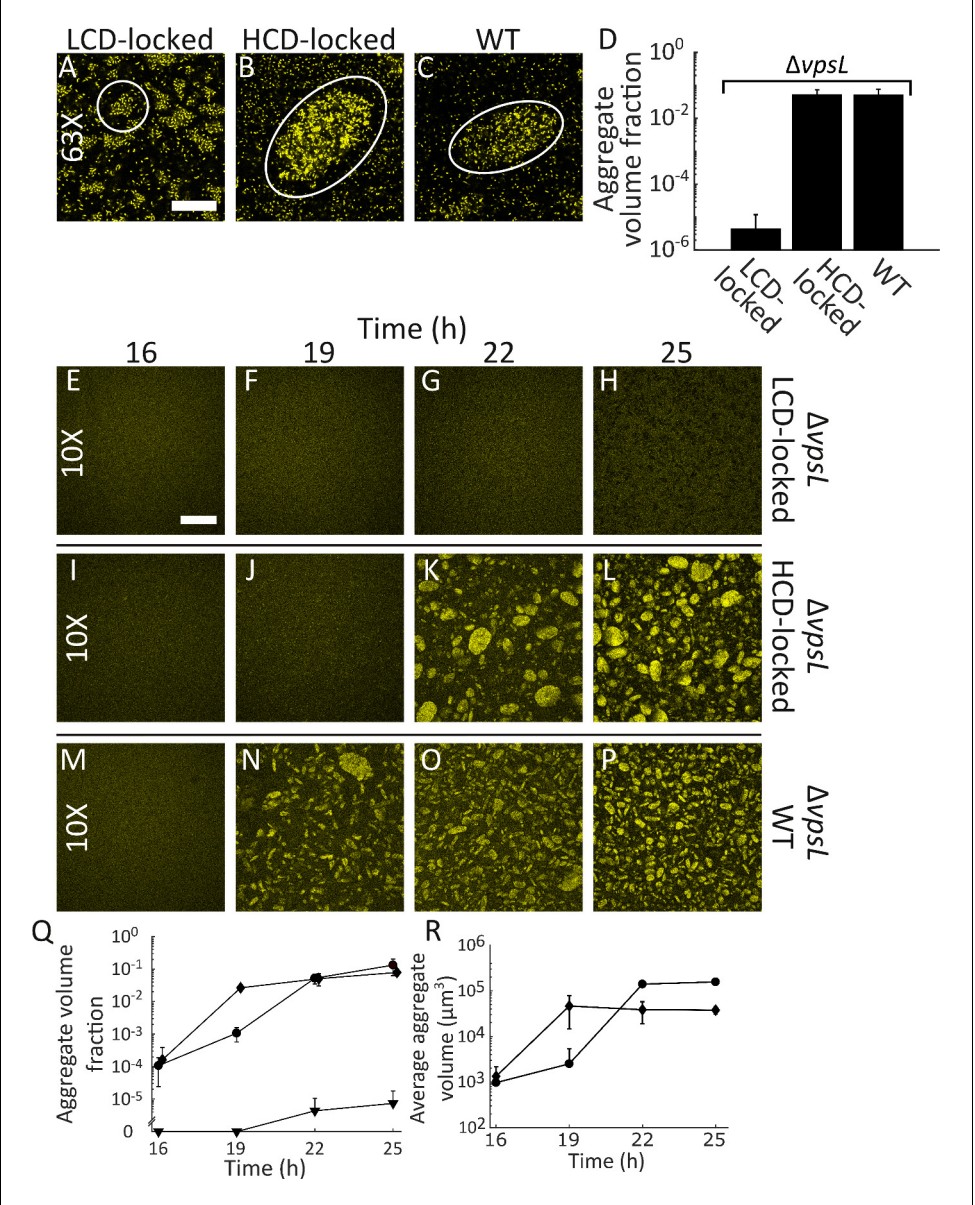

**Figure 2.** Quorum sensing controls rapid, VpsL-independent aggregation of *V. cholerae* in liquid. Aggregate formation of the LCD QS-locked (LCD-locked) (**A**), HCD QS-locked (HCD-locked) (**B**), and wild-type (WT) (**C**) *V. cholerae* strains after 22 h of growth. Shown are representative cross-sections through samples (**A–C**). The approximate extents of individual aggregates are indicated with white outlines. Magnification: 63X; scale bar: 50 μM. (**D**) Quantitation of total volume fraction, the total volume of the imaged region that is occupied by aggregates (Materials and methods) within the imaged region for Δ*vpsL* LCD-locked, Δ*vpsL* HCD-locked, and Δ*vpsL* WT strains after 22 h of growth. Representative cross-sections through the Δ*vpsL* LCD-locked (**E–H**), Δ*vpsL* HCD-locked (**I–L**), and Δ*vpsL* WT (**M–P**) *V. cholerae* strains at 16, 19, 22, and 25 h. (**E–P**) Magnification: 10X; scale bar: 250 μm. (**Q**) Quantitation of aggregate volume fraction. The data for T = 22 h are the same as those shown in *Figure 2D*. Triangle: Δ*vpsL* LCD-locked, circle: Δ*vpsL* HCD-locked, and diamond: Δ*vpsL* WT. (**R**) Average cluster volume over time for the Δ*vpsL* HCD-locked (circle) and Δ*vpsL* WT (diamond) strains. (**D,Q,R**) Quantitation of mean ± standard deviation (SD; N=3 biological replicates). Mean and SD were calculated using the untransformed data, not the log-transformed data, which results in asymmetric error bars. All strains in all panels harbor the fluorescent *mKO* reporter constitutively expressed from the chromosome.

DOI: https://doi.org/10.7554/eLife.42057.003

The following source data and figure supplements are available for figure 2:

**Source data 1.** Figure source data.

*Figure 2 continued on next page*

*Figure 2 continued*

DOI: https://doi.org/10.7554/eLife.42057.014

**Figure supplement 1.** *V. cholerae* aggregates form in liquid.

DOI: https://doi.org/10.7554/eLife.42057.004

**Figure supplement 2.** *V. cholerae* forms distinct aggregates.

DOI: https://doi.org/10.7554/eLife.42057.005

**Figure supplement 3.** Known $Ca^{2+}$-related genes do not contribute to *V. cholerae* aggregation.

DOI: https://doi.org/10.7554/eLife.42057.006

**Figure supplement 3—source data 1.** Figure source data.

DOI: https://doi.org/10.7554/eLife.42057.007

**Figure supplement 4.** Known pili genes do not contribute to *V. cholerae* aggregation.

DOI: https://doi.org/10.7554/eLife.42057.008

**Figure supplement 4—source data 1.** Figure source data.

DOI: https://doi.org/10.7554/eLife.42057.009

**Figure supplement 5.** WT and Δ*vpsL* strains display similar aggregation kinetics.

DOI: https://doi.org/10.7554/eLife.42057.010

**Figure supplement 6.** *V. cholerae* aggregate formation is rapid.

DOI: https://doi.org/10.7554/eLife.42057.011

**Figure supplement 6—source data 1.** Figure source data.

DOI: https://doi.org/10.7554/eLife.42057.012

**Figure supplement 7.** *V. cholerae* aggregate formation is non-clonal.

DOI: https://doi.org/10.7554/eLife.42057.013

strain and there was up to a 3 h delay in aggregation for the Δ*vpsL* HCD-locked strain. In both cases, once initiated, aggregation occurred rapidly before reaching a steady state (*Figure 2J–L,N–P*). Indeed, quantitation shows that, following the rapid onset of aggregation (at T = 19 h for the Δ*vpsL* WT strain and at T = 22 h for the Δ*vpsL* HCD-locked strain), only modest changes occur in volume fraction occupied by aggregates (*Figure 2Q*) and in aggregate size distribution (*Figure 2R*) from T = 22–25 h. As a control, we show that WT cells exhibit similar aggregation kinetics as the Δ*vpsL* WT strain (*Figure 2—figure supplement 5*).

To more precisely define the timing of aggregate development, we imaged the Δ*vpsL* HCD-locked strain every 30 min during the key 19–22 h window (*Figure 2—figure supplement 6*). We found that onset to completion of aggregation occurs within a 30 min period. Importantly, the rapidity of aggregate formation precludes a process in which cell division drives aggregate growth. Rather, aggregation must be driven by a change that causes existing cells to adhere and form aggregates. In support of this view, growth of a mixture of equal numbers of the Δ*vpsL* HCD-locked strain constitutively expressing either *mKate2* or *mTFP1* resulted in aggregates containing both red- and teal-colored cells (*Figure 2—figure supplement 7*). Again, in line with our argument that this is a new community formation process, the non-clonal nature of aggregate formation is distinct from the well-established, strictly-clonal *V. cholerae* surface-biofilm formation program (*Nadell et al., 2015*).

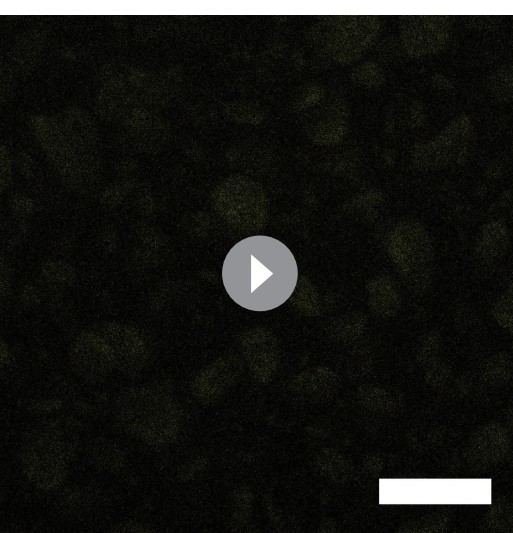

**Video 1.** Δ*vpsL* HCD-locked *V. cholerae* aggregates. z-scan through a representative sample of aggregates formed by a Δ*vpsL* HCD-locked strain at T = 22 h. Data are the same as shown in *Figure 2K*. Magnification: 10X; scale bar: 250 μm. Strain harbors *mKO* constitutively expressed from the chromosome.

DOI: https://doi.org/10.7554/eLife.42057.015

## Exogenous supplementation of QS autoinducers induces aggregation

Our finding that aggregation occurs in the Δ*vpsL* HCD-locked strain but not in the Δ*vpsL* LCD-locked strain implies that the process is QS-controlled. Thus, we predicted that the accumulation of QS autoinducers, which occurs as *V. cholerae* naturally transitions from the LCD to the HCD QS-state, should induce aggregation. To test this prediction, we required strains that only detect one specific autoinducer and that cannot produce that autoinducer. In such strains, the autoinducer can be supplied exogenously and the response quantified. Based on this logic, we constructed two different strains (see *Figure 1*). One strain lacks the CAI-1 synthase, CqsA, and the AI-2 receptor, LuxQ, and thus responds exclusively to CAI-1. The second strain lacks the AI-2 synthase, LuxS, and the CAI-1 receptor, CqsS, and thus responds exclusively to AI-2. As noted in the Introduction, *V. cholerae* has two other QS receptors, VpsS and CqsR, which funnel information into the QS circuit, and which detect unknown autoinducers (*Jung et al., 2015*; *Shikuma et al., 2009*). To ensure that none of the effects we measured is a consequence of the two unknown ligands, we deleted the *cqsR* and *vpsS* genes encoding their receptors. Our CAI-1-responsive strain is Δ*cqsA* Δ*luxQ* Δ*vpsS* Δ*cqsR* Δ*vpsL*. Our AI-2-responsive strain is Δ*luxS* Δ*cqsS* Δ*vpsS* Δ*cqsR* Δ*vpsL*.

At T = 0 h, we added CAI-1 (5 μM) to the CAI-1-responsive strain and AI-2 (1 μM together with 100 μM boric acid) to the AI-2-responsive strain. These are saturating concentrations for each autoinducer. We assessed aggregation formation at 22 h. Administration of either CAI-1 or AI-2 to the respective autoinducer-responsive strain drove aggregation, whereas no aggregation occurred in samples to which a solvent control was supplied (*Figure 3A*). Additionally, autoinducer driven aggregation should require the cognate receptor to be present to transduce the QS information into the cell. Indeed, the autoinducers had no effect on aggregation if the strain to which CAI-1 or AI-2 was added lacked CqsS or LuxQ, respectively (*Figure 3—figure supplement 1*). Lastly, the presence or absence of *vpsL*, *vpsS*, and *cqsR* did not influence responsiveness to exogenously supplied autoinducers (*Figure 3—figure supplement 2*).

To define the temporal response window for autoinducers to trigger aggregation, we used the above CAI-1-responsive strain as our test case. We focused on the role of CAI-1 because it is the stronger of the two *V. cholerae* autoinducers (*Miller et al., 2002*). Pilot experiments showed that CAI-1 supplementation at or after 16 h could not promote aggregation. Thus, we supplemented cultures with 5 μM CAI-1 at one-hour intervals from 3 to 8 h. At T = 22 h, we measured aggregation (*Figure 3B*, black bars). CAI-1 addition between 3–6 h promoted aggregation, whereas from 7 h onward, CAI-1 had no effect on aggregation. Addition of CAI-1 at late times did not simply delay the onset of aggregation: at T = 46 h, cultures to which CAI-1 had been added at 7–8 h still showed no aggregation (*Figure 3—figure supplement 3*). Importantly, cells in our analyses remain capable of detecting and controlling established QS-regulated genes in response to late-time CAI-1 supplementation. Specifically, the CAI-1-responsive strain harboring the QS-controlled luciferase genes produced maximal light at T = 22 h, independent of the time point (0 h, 3–8 h) at which CAI-1 was supplied (*Figure 3B*, gray bars). Combined, these data show that extracellular autoinducers trigger aggregation in *V. cholerae*, albeit only within a specific temporal window, while canonical autoinducer-triggered QS-behaviors are not subject to such temporal restriction. Under our growth conditions, T = 7 h, after which the aggregation phenotype becomes impervious to autoinducer supplementation, roughly corresponds to when the cells enter stationary phase. We consider stationary-phase-specific factors that could play roles in aggregation in the Discussion.

## Aggregation is HapR dependent

The architecture of the *V. cholerae* QS circuit predicts three possible mechanisms by which QS could promote aggregation at HCD: AphA could repress aggregation at LCD, HapR could promote aggregation at HCD, or the Qrr sRNAs could control aggregation independently of AphA and HapR (*Figure 1*). To define the QS path to aggregate formation, we constructed strains lacking *hapR* and/or *aphA* in both the Δ*vpsL* LCD-locked and Δ*vpsL* HCD-locked strains and measured aggregation at 22 h.

There was no difference in aggregation between the Δ*vpsL* HCD-locked strain and the Δ*vpsL* Δ*aphA* HCD-locked strain (*Figure 3C,D,K*). Thus, AphA does not control aggregate formation, eliminating the first possibility. By contrast, deletion of *hapR* in the Δ*vpsL* HCD-locked strain led to a complete loss of aggregation (*Figure 3E,K*), with levels of aggregation comparable to a Δ*vpsL* LCD-

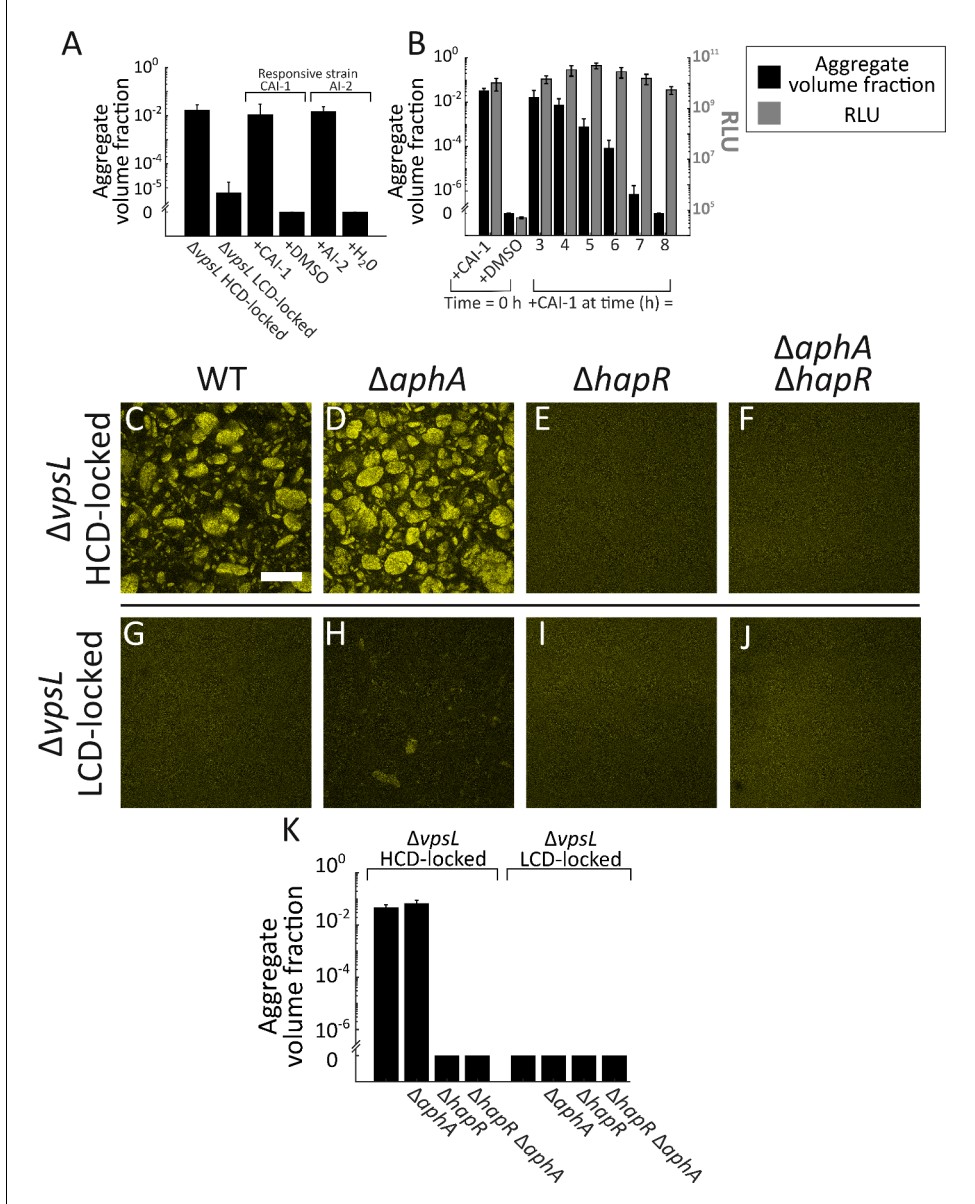

**Figure 3.** Exogenous autoinducers drive *V. cholerae* aggregation and HapR is required. Quantitation of aggregate volume fraction at 22 h for the Δ*vpsL* HCD QS-locked (HCD-locked), Δ*vpsL* LCD QS-locked (LCD-locked), CAI-1-responsive (± CAI-1), and AI-2-responsive (± AI-2 and boric acid) *V. cholerae* strains (**A**). Autoinducers or solvent controls were added at the time of inoculation. Concentrations used: CAI-1: 5 μM, AI-2: 1 μM, and boric acid: 100 μM. (**B**) Quantitation of aggregate volume fraction (black bars) and bioluminescence (gray bars) at 22 h for the CAI-1-responsive strain to which CAI-1 was added at T = 0 h and from 3 to 8 h at 1 h intervals. Also shown is bioluminescence quantified in a CAI-1-responsive strain harboring the cosmid pBB1 which carries the *luxCDABE* genes. RLU denotes relative lights units, defined as counts/min mL$^{-1}$ per OD$_{600}$. In **A** and **B** aggregate volume fraction was quantified in a strain harboring *mKO* constitutively expressed from the chromosome; quantitation of mean ± standard deviation (SD) (N=3 biological replicates). Representative cross-sections of the Δ*vpsL* HCD-locked (**C**), Δ*vpsL* Δ*aphA* HCD-locked (**D**), Δ*vpsL* Δ*hapR* HCD-locked (**E**), Δ*vpsL* Δ*aphA* Δ*hapR* HCD-locked (**F**), Δ*vpsL* LCD-locked (**G**), Δ*vpsL* Δ*aphA* LCD-locked (**H**), Δ*vpsL* Δ*hapR* LCD-locked (**I**), and Δ*vpsL* Δ*aphA* Δ*hapR* LCD-locked (**J**) *V. cholerae* strains following 22 h of growth. (**C–J**) Magnification: 10X; scale bar: 250 μm. All strains harbor *mKO* constitutively expressed from the chromosome. (**K**) Quantitation of aggregate volume fraction for samples in **C–J**. Shown are mean ± SD (N=3 biological replicates). The Δ*vpsL* Δ*aphA* LCD-locked strain appears to exhibit modest aggregation (**H**), possibly due to AphA repression of *hapR* transcription (*Rutherford et al., 2011*), but the level of aggregation is below the detection threshold employed in the segmenting analysis (**K**).

*Figure 3 continued on next page*

*Figure 3 continued*

DOI: https://doi.org/10.7554/eLife.42057.016

The following source data and figure supplements are available for figure 3:

**Source data 1.** Figure source data.
DOI: https://doi.org/10.7554/eLife.42057.025
**Figure supplement 1.** Autoinducer supplementation drives *V. cholerae* aggregation via the cognate QS receptor.
DOI: https://doi.org/10.7554/eLife.42057.017
**Figure supplement 1—source data 1.** Figure source data.
DOI: https://doi.org/10.7554/eLife.42057.018
**Figure supplement 2.** Autoinducer supplementation drives *V. cholerae* aggregation in the presence of *vpsL, cqsR*, and *vpsS*.
DOI: https://doi.org/10.7554/eLife.42057.019
**Figure supplement 2—source data 1.** Figure source data.
DOI: https://doi.org/10.7554/eLife.42057.020
**Figure supplement 3.** Late-time autoinducer supplementation does not delay the onset of aggregation.
DOI: https://doi.org/10.7554/eLife.42057.021
**Figure supplement 3—source data 1.** Figure source data.
DOI: https://doi.org/10.7554/eLife.42057.022
**Figure supplement 4.** Complementation of *hapR* in aggregate formation.
DOI: https://doi.org/10.7554/eLife.42057.023
**Figure supplement 4—source data 1.** Figure source data.
DOI: https://doi.org/10.7554/eLife.42057.024

locked strain (*Figure 3G,K*). Thus, HapR is required for aggregation, showing that the second possibility is correct. Indeed, epistasis analysis demonstrates that the phenotype of the Δ*vpsL* Δ*aphA* Δ*hapR* HCD-locked mutant strain was identical to that of the Δ*vpsL* Δ*hapR* HCD-locked mutant (*Figure 3E,F,K*).

We performed the analogous experiments in the Δ*vpsL* LCD-locked set of strains. When *hapR* was absent, no aggregation occurred (*Figure 3I,K*). We note that deletion of *aphA* in the Δ*vpsL* LCD-locked strain led to a modest increase in aggregation (*Figure 3H*), but one that remains below the threshold for detection by our segmentation protocol (*Figure 3K*). AphA represses transcription of *hapR* (*Rutherford et al., 2011*). Thus, the Δ*vpsL* Δ*aphA* LCD-locked strain has elevated HapR levels, and our genetic analysis above shows that HapR promotes aggregation. AphA regulation of *hapR* could account for the minor increase in aggregation evident in *Figure 3H*. Aggregation does not occur in the Δ*vpsL* Δ*aphA* Δ*hapR* LCD-locked strain (*Figure 3J,K*), which is consistent with any AphA regulation of aggregation occurring through HapR. To confirm the role of HapR in aggregation, we complemented the Δ*vpsL* Δ*hapR* HCD-locked strain with a chromosomal copy of *hapR* that we introduced at the *lacZ* locus. At T = 22 h, the Δ*vpsL* Δ*hapR* *lacZ:P*<sub>hapR</sub>-*hapR* HCD-locked strain exhibited a comparable level of aggregation to the Δ*vpsL* HCD-locked strain (*Figure 3—figure supplement 4*). We conclude that HapR is the main QS activator of aggregation. Above, we mentioned that a third regulatory possibility was that the Qrr sRNAs control aggregation independently of AphA and HapR. Our experiments here show that HapR is essential for aggregation, eliminating this final option.

## eDNA contributes to aggregation

To determine structural components of aggregates, we first took a candidate approach using the logic that genes encoding such components must be regulated by HapR at HCD. A promising candidate is Dns, one of two *V. cholerae* extracellular nucleases. HapR represses *dns* at HCD (*Blokesch and Schoolnik, 2008*). The other extracellular nuclease, Xds, is not known to be HapR-controlled. Both nucleases contribute to the surface-biofilm program (*Seper et al., 2011*). We reasoned that at HCD, HapR repression of *dns*, coupled with low or no Xds activity, would promote eDNA production potentially contributing to aggregation.

To explore the role of eDNA in aggregate formation, we constructed mutants lacking both *xds* and *dns* in Δ*vpsL* LCD-locked and Δ*vpsL* HCD-locked strains. We analyzed these two strains, along with the parent Δ*vpsL* LCD-locked and Δ*vpsL* HCD-locked strains over time. At 16 and 19 h, as

discussed above, the ΔvpsL HCD-locked strain consisted primarily of planktonic cells and a small number of clusters (*Figure 4A*). All other strains consisted of exclusively planktonic cells (*Figure 4A*). At 22 h, both the ΔvpsL HCD-locked and the ΔvpsL Δxds Δdns HCD-locked strains formed aggregates (*Figure 4A,B,D*), however the average aggregate size was larger in the ΔvpsL Δxds Δdns HCD-locked strain than in the ΔvpsL HCD-locked strain (*Figure 4F*). Because the ΔvpsL Δxds Δdns HCD-locked strain formed aggregates that were larger than the imaging field of view used for the above experiments, in panel 4F, we estimated the aggregate size of these strains by computing the cross-sectional area of all aggregates within an entire microtiter dish. The ΔvpsL LCD-locked and ΔvpsL Δxds Δdns LCD-locked strains exhibited no aggregation (*Figure 4A,C,E*). Individual deletion of *xds* or *dns* in the ΔvpsL HCD-locked strain also resulted in aggregation (*Figure 4—figure supplement 1A*), with Dns contributing more than Xds to overall aggregate size (*Figure 4—figure supplement 1B*). These relative effects parallel those reported for Xds and Dns in surface-biofilm formation (*Seper et al., 2011*). Unlike for the ΔvpsL HCD-locked strain, aggregates of the ΔvpsL Δxds Δdns HCD-locked strain continued to enlarge, precluding an accurate assessment of aggregate size after T = 22 h. To verify the role of extracellular nucleases, we complemented the ΔvpsL Δxds Δdns HCD-locked strain by introducing a chromosomal copy of *dns* at the *lacZ* locus to restore the stronger of the two extracellular nucleases. The ΔvpsL Δxds Δdns lacZ:$P_{dns}$-dns HCD-locked strain produced smaller aggregates than the ΔvpsL Δxds Δdns HCD-locked strain (*Figure 4—figure supplement 2*). We conclude that extracellular nucleases function to limit overall aggregate size but they do not control aggregation timing.

Curiously, no aggregation occurred in the ΔvpsL Δxds Δdns LCD-locked strain. We expected eDNA levels to be elevated in this strain because it lacks extracellular nucleases responsible for eDNA degradation, and, based on the above results with the ΔvpsL Δxds Δdns HCD-locked strain, the presence of eDNA influences aggregation. We measured the eDNA content in all four strains at 22 h, and indeed, the concentrations of eDNA in the ΔvpsL Δxds Δdns LCD-locked and ΔvpsL Δxds Δdns HCD-locked strains are equivalent and, moreover, elevated ~4-fold and 10-fold compared to the ΔvpsL HCD-locked and ΔvpsL LCD-locked strains, respectively (*Figure 4G*). These data show that, by itself, accumulation of eDNA is not sufficient to drive aggregation.

We confirmed that eDNA is present in aggregates by imaging the ΔvpsL HCD-locked, and ΔvpsL Δxds Δdns HCD-locked strains grown to 22 h followed by addition of the cell-impermeant nucleic acid stain TOTO-1. In the ΔvpsL HCD-locked strain, patches of eDNA were present in aggregates (*Figure 4H*, arrow; *Video 2*) and a stronger eDNA signal could be visualized in the ΔvpsL Δxds Δdns HCD-locked strain (*Figure 4I*, arrow; *Video 3*). In both strains, we also identified dead cells (*Figure 4H,I*; *Videos 2–3*). To further test the contribution of eDNA, we supplied 100 Kunitz units/mL DNase I to strains at T = 0 h. Because the quantity of DNAse I required for experiments such as those in *Figure 4A*, which are in carried out in 2 mL volumes, is prohibitive, we modified our protocol for this analysis to enable use of small volumes. DNase I treatment reduced aggregation in the ΔvpsL Δxds Δdns HCD-locked strain but had little effect on the ΔvpsL HCD-locked strain (*Figure 4—figure supplement 3*). The timing of aggregation was altered in microtiter-dish-grown cells, hence the difference in the assayed time point. This result supports the conclusion from above that eDNA contributes to, but is not sufficient for, aggregate formation. Combined, the data in *Figure 4* show that eDNA levels are modulated through the activity of two extracellular nucleases, and eDNA, in turn, plays a role in *V. cholerae* aggregate formation in the HCD QS-mode. However, the lack of aggregate formation in the ΔvpsL Δxds Δdns LCD-locked strain, in the face of elevated eDNA levels, coupled with the finding that complete loss of aggregation does not occur upon complementation with *dns* or exogenous DNase I supplementation, argues that there must be additional components required for *V. cholerae* aggregate formation.

## A genetic screen to identify components required for *V. cholerae* aggregation

We developed a screen to uncover genes involved in *V. cholerae* aggregate formation by exploiting a readily assayable plate-based phenotype that correlated with aggregation in liquid. On agar plates, ΔvpsL HCD-locked and ΔvpsL LCD-locked colonies formed opaque and translucent colonies, respectively (*Figure 5A,B*). Variability in *V. cholerae* colony opacity has been previously reported (*Finkelstein et al., 1992*; *Finkelstein et al., 1997*; *Lankford, 1960*), but, to our knowledge, has not been linked to aggregation. Analogous colony phenotypes are well known in other species of *Vibrio*,

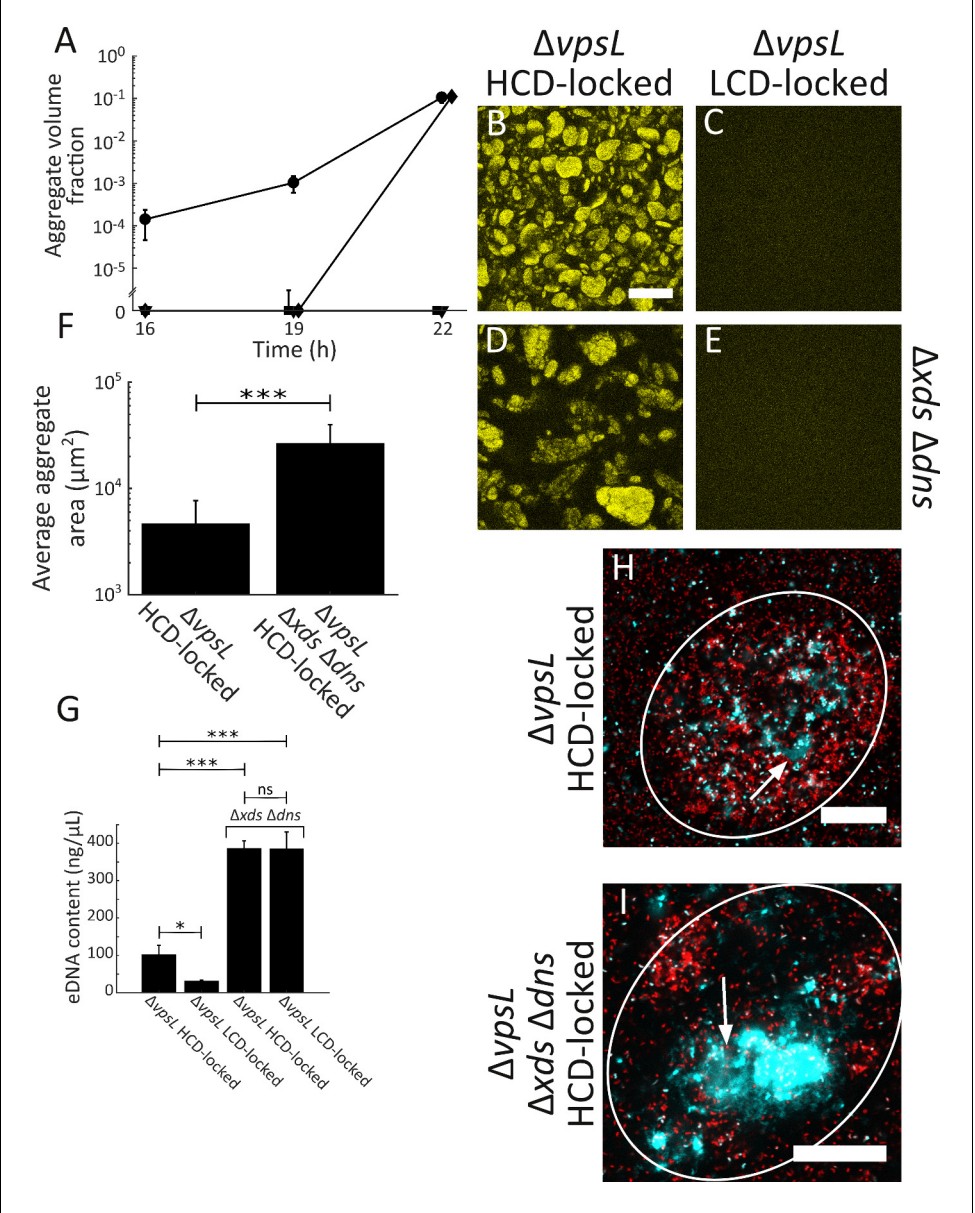

**Figure 4.** Extracellular DNA contributes to *V. cholerae* aggregation. (**A**) Quantitation of aggregate volume fraction over time. Circle: Δ*vpsL* HCD QS-locked (HCD-locked), triangle: Δ*vpsL* LCD QS-locked (LCD-locked), diamond: Δ*vpsL* Δ*xds* Δ*dns* HCD-locked, and square: Δ*vpsL* Δ*xds* Δ*dns* LCD-locked. Representative cross-sections of the Δ*vpsL* HCD-locked (**B**), Δ*vpsL* LCD-locked (**C**), Δ*vpsL* Δ*xds* Δ*dns* HCD-locked (**D**), Δ*vpsL* Δ*xds* Δ*dns* LCD-locked (**E**) *V. cholerae* strains following 22 h of growth. Magnification: 10X; scale bar: 250 μm. (**F**) Average aggregate cross-sectional area at T = 22 h for the Δ*vpsL* HCD-locked and Δ*vpsL* Δ*xds* Δ*dns* HCD-locked strains. Statistical significance was determined with a two-sample Kolmogorov–Smirnov test on pooled data (***= p<0.0005). (**A–F**) All strains harbor *mKO* constitutively expressed from the chromosome. (**G**) Quantitation of total bulk eDNA content in Δ*vpsL* HCD-locked, Δ*vpsL* LCD-locked, Δ*vpsL* Δ*xds* Δ*dns* HCD-locked, and Δ*vpsL* Δ*xds* Δ*dns* LCD-locked strains following 22 h of growth. Statistical significance was determined with a two-sample t-test (*= p<0.05, ***= p<0.0005, ns = not significant). (**H**) Cross-section through a representative culture of the Δ*vpsL* HCD-locked strain (red) to which the eDNA stain TOTO-1 (cyan) was added following 22 h of growth. (**I**) Cross-section through a representative culture of the Δ*vpsL* Δ*xds* Δ*dns* HCD-locked strain (red) to which the eDNA stain TOTO-1 (cyan) was added following 22 h of growth. In (**H,I**), white arrows indicate regions of eDNA. Strains harbor *mKate2* constitutively expressed from the chromosome. Magnification: 63X; scale bar: 25 μm. Samples are representative of 3 biological replicates. (**A,F,G**) Quantitation of mean ± standard deviation (N=3 biological replicates).

*Figure 4 continued on next page*

*Figure 4 continued*

DOI: https://doi.org/10.7554/eLife.42057.026

The following source data and figure supplements are available for figure 4:

**Source data 1.** Figure source data.
DOI: https://doi.org/10.7554/eLife.42057.033

**Figure supplement 1.** *V. cholerae* aggregate size is controlled by both Xds and Dns.
DOI: https://doi.org/10.7554/eLife.42057.027

**Figure supplement 1—source data 1.** Figure source data.
DOI: https://doi.org/10.7554/eLife.42057.028

**Figure supplement 2.** Complementation of *dns* in aggregate formation.
DOI: https://doi.org/10.7554/eLife.42057.029

**Figure supplement 2—source data 1.** Figure source data.
DOI: https://doi.org/10.7554/eLife.42057.030

**Figure supplement 3.** DNase I supplementation reduces aggregate size in the Δ*vpsL* Δ*xds* Δ*dns* HCD-locked strain.
DOI: https://doi.org/10.7554/eLife.42057.031

**Figure supplement 3—source data 1.** Figure source data.
DOI: https://doi.org/10.7554/eLife.42057.032

including *V. parahaemolyticus* and *V. alginolyticus,* and in those organisms, opacity/translucence are QS-regulated and related to capsule synthesis (*Chang et al., 2009*; *Enos-Berlage and McCarter, 2000*; *Enos-Berlage et al., 2005*). Germane to our study is that the Δ*vpsL* HCD-locked *V. cholerae* strain, which is proficient for aggregation, forms opaque colonies, while deletion of *hapR* results in translucent colonies, which correlates with the loss of the ability to aggregate (*Figure 5C*).

We reasoned that mutagenesis of the Δ*vpsL* HCD-locked *V. cholerae* strain followed by screening for translucent colonies could reveal genes involved in aggregation. We used Tn*5* to randomly mutagenize the Δ*vpsL* HCD-locked strain as well as a

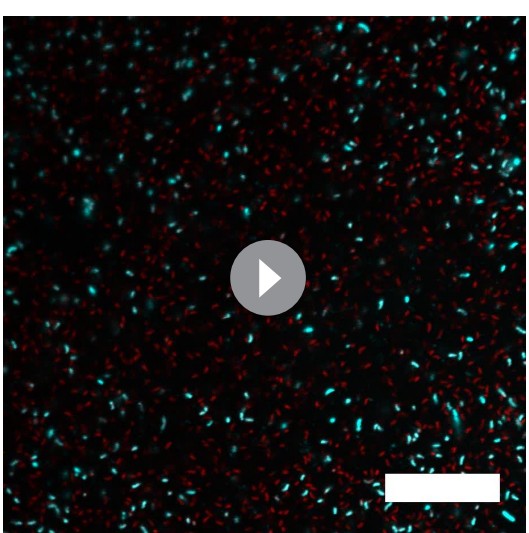

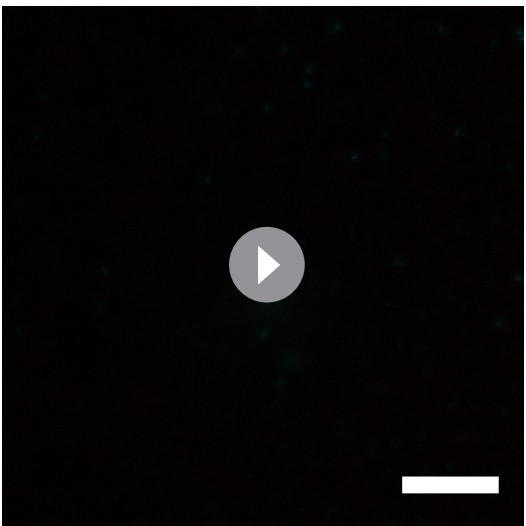

**Video 2.** eDNA in the Δ*vpsL* HCD-locked *V. cholerae* aggregate. z-scan through a representative aggregate of the Δ*vpsL* HCD-locked strain (red) to which the eDNA stain TOTO-1 (cyan) was added following 22 h of growth. The strain harbors *mKate2* constitutively expressed from the chromosome. Magnification: 63X; scale bar: 25 μm. Contrast independently adjusted in *Videos 2* and *3* to highlight different eDNA features.
DOI: https://doi.org/10.7554/eLife.42057.034

**Video 3.** eDNA in the Δ*vpsL* Δ*xds* Δ*dns* HCD-locked *V. cholerae* aggregate. z-scan through a representative aggregate of the Δ*vpsL* Δ*xds* Δ*dns* HCD-locked strain (red) to which the eDNA stain TOTO-1 (cyan) was added following 22 h of growth. The strain harbors *mKate2* constitutively expressed from the chromosome. Magnification: 63X; scale bar: 25 μm. Contrast independently adjusted in *Videos 2* and *3* to highlight different eDNA features.
DOI: https://doi.org/10.7554/eLife.42057.035

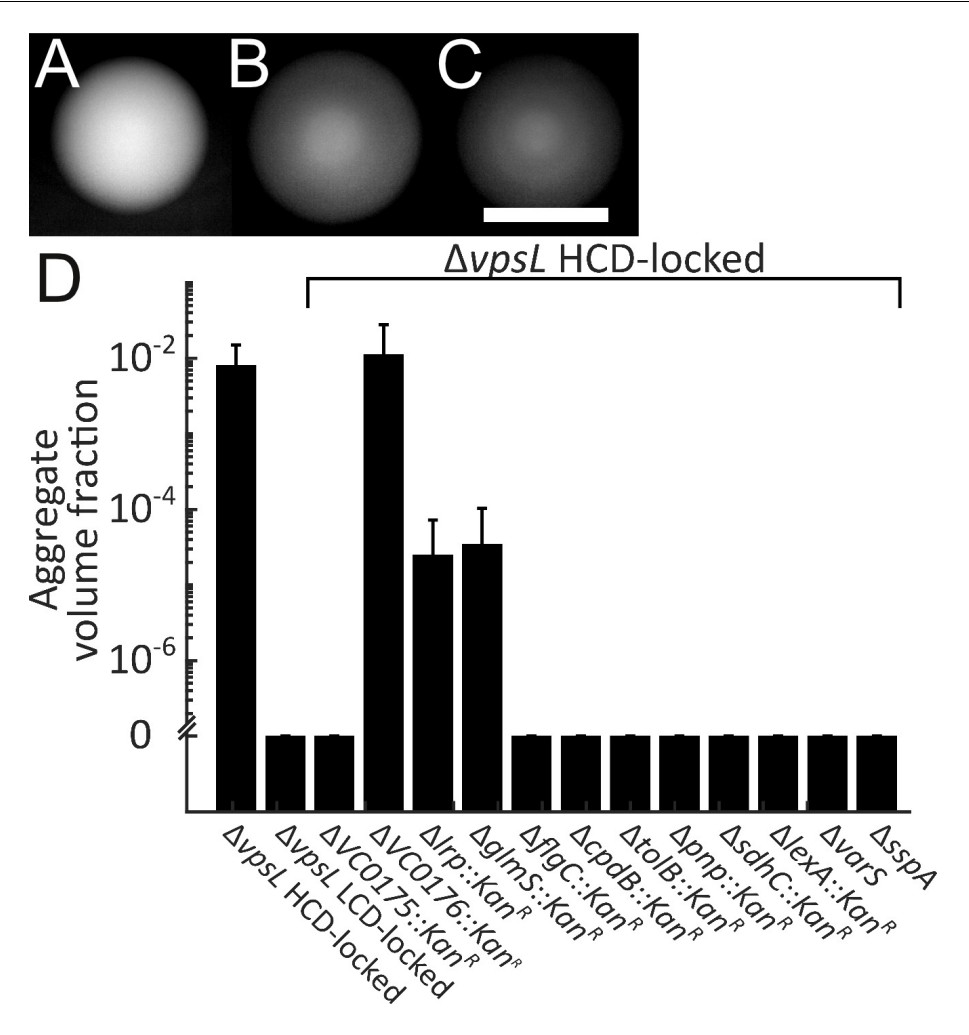

**Figure 5.** Genes required for *V. cholerae* aggregation. Representative Δ*vpsL* HCD QS-locked (HCD-locked) (**A**, opaque), Δ*vpsL* LCD QS-locked (LCD-locked) (**B**, translucent), and Δ*vpsL* Δ*hapR* HCD-locked (**C**, translucent) colonies grown on LB agar plates at 37°C for 24 h. (**A–C**) Scale bar: 5 mm. (**D**) Quantitation of aggregate volume fraction for Δ*vpsL* HCD-locked, Δ*vpsL* LCD-locked, and Δ*vpsL* HCD-locked strains carrying deletions in each of the genes identified in the screen (***Table 1***). Samples were stained with the nucleic acid stain SYTO-9. Quantitation of mean ± standard deviation (N≥3 biological replicates) after 22 h of growth.
DOI: https://doi.org/10.7554/eLife.42057.036

The following source data is available for figure 5:

**Source data 1.** Figure source data.
DOI: https://doi.org/10.7554/eLife.42057.037

---

Δ*vpsL lacZ:P*$_{hapR}$-*hapR* HCD-locked strain. We used the *hapR* merodiploid to avoid identifying insertions in *hapR*, which we knew would cause a translucent phenotype. Our rationale for also screening in the strain with only a single copy of *hapR* was because, although we were attracted to the idea of eliminating *hapR* mutants, we were concerned that as QS is involved in aggregate formation, overproduction of HapR could mask potential phenotypes. We screened ~25 000 mutants in each case. Both screens were successful and yielded overlapping sets of genes. In the strain containing only a single copy of *hapR*, we used polymerase chain reaction (PCR) to identify and eliminate from analysis mutants with transposon insertions in *hapR*. Following this procedure, we identified a total of 49 colonies exhibiting translucent phenotypes (from both screens). We successfully identified the transposon insertion locations in 45 of the mutants, revealing 18 unique loci. We carried out a secondary screen to determine whether disruption of the candidate genes, beyond conferring the translucent

**Table 1.** Genes that contribute to aggregation in *V. cholerae*

| Gene locus | Annotation | *hapR* merodiploid/haploid |
| --- | --- | --- |
| *vc0092* | LexA transcriptional repressor (*lexA*) | Merodiploid |
| *vc0175* | Deoxycytidylate deaminase-like protein, putative | Merodiploid |
| *vc0487* | Glucosamine-fructose-6-phosphate aminotransferase (*glmS*) | Merodiploid |
| *vc0576* | Stringent starvation protein A (*sspA*) | Merodiploid |
| *vc0647* | Polyribonucleotide nucleotidyltransferase (*pnp*) | Both |
| *vc1836* | Translocation protein (*tolB*) | Haploid |
| *vc1904* | Leucine-responsive transcriptional regulator (*lrpA*) | Merodiploid |
| *vc2091-vc2087* | Succinate and 2-oxoglutarate dehydrogenases (TCA cycle) (*sdhC, sdhD, sdhA, sdhB, and sucA*) | Both |
| *vc2200-vc2198* | Flagellar basal-body rod proteins (*flgBCD*) | Both |
| *vc2453* | Hybrid sensor histidine kinase VarS (*varS*) | Merodiploid |
| *vc2562* | 2'3'cyclic phosphodiesterase (*cpdB*) | Haploid |

DOI: https://doi.org/10.7554/eLife.42057.038

phenotype, altered aggregation in liquid. Mutations in 12 of the 18 identified loci led to strains displaying greatly diminished aggregation (*Table 1* reports all of the genes and from which screen they were obtained). We validated these observations by engineering deletions of the candidate genes in the Δ*vpsL* HCD-locked strain and assaying for defects in aggregation at 22 h, a time point by which the Δ*vpsL* HCD-locked parent strain consistently formed aggregates. We confirmed that aggregation is diminished or eliminated in all of the mutant strains (*Figure 5D*). We identified multiple insertions in the operons encoding components of the TCA cycle – the succinate and 2-oxoglutarate dehydrogenase complexes (*vc2091-vc2087; sdhC, sdhD, sdhA, sdhB, and sucA*) and the flagellar basal body (*vc2200-vc2198; flgBCD*). We deleted one representative gene (*sdhC* and *flgC*) to confirm the phenotype. Additionally, two of the insertions identified in our screen are located in intergenic regions between divergently transcribed operons (*vca0125/vca0127* and *vca0175/vca0176*). Our attempts to delete the *vca0125/vca0127* region were unsuccessful so we could not verify the role in aggregation. In the case of the *vca0175/vca0176* region, we individually deleted *vca0175* and *vca0176*, and determined that *vca0175*, which has no known function, is the gene responsible for the aggregation defect. While investigating the mechanisms by which the genes identified in our screen regulate aggregation is beyond the scope of this work, we lay out possible roles below.

## Discussion

Here, we demonstrate the existence of an aggregation process in *V. cholerae* that is independent of the known surface-biofilm program. Aggregation occurs in liquid and is rapid, suggesting that cell division is not required. Aggregate formation occurs when *V. cholerae* cells are in a HCD QS-state, a QS-regulation pattern opposite to that for surface-biofilm formation which occurs when cells are in a LCD QS-state (*Hammer and Bassler, 2003*; *Zhu and Mekalanos, 2003*). Aggregate formation is promoted by exogenous autoinducers, although only during a limited temporal window. HapR, the master regulator of HCD QS-behavior, is required for aggregate formation. eDNA is present in aggregates, contributes to overall aggregate size, and two extracellular nucleases, Xds and Dns, are involved. eDNA is not, however, sufficient to drive aggregate formation. For aggregate formation to

occur, genes involved in stress response, phosphate uptake from eDNA, genes of unknown function (including a gene, *vc0175*, that distinguishes the current seventh pandemic El Tor biotype from the previous classical biotype) are required. The identification of *cpdB*, which is involved in phosphate acquisition from eDNA (*McDonough et al., 2016*), and our demonstration of Xds- and Dns-driven regulation of eDNA production in aggregates, suggests that eDNA acquisition from the environment may be important for aggregate-associated cells.

There is a growing recognition that bacteria form multicellular aggregates in liquid, a state that, relative to individual planktonic cells, can confer fitness benefits including increased antibiotic resistance and improved surface-colonization relative to planktonic cells (*Kragh et al., 2016*; *Kragh et al., 2018*; *Schleheck et al., 2009*). Bacterial aggregation can be modulated by factors including QS-state, eDNA, ions, and cationic polymers (*Chandler et al., 2009*; *Das et al., 2014*; *Laganenka et al., 2016*; *Perez-Soto et al., 2018*). Combined, these findings begin to argue that bacteria exhibit multicellular behaviors in the liquid-phase that are not captured by studies of surface-bound bacterial communities.

We propose a model for how aggregate formation could be instrumental in the two major *V. cholerae* habitats—the human host and the marine environment—as well as during transitions between them. We consider each niche in turn. First, *V. cholerae* aggregation during infection: formation of multicellular communities occurs during human infection (*Teschler et al., 2015*). Filtration that removes aggregates and copepod-associated bacteria, but not planktonic cells, reduced the efficacy of *V. cholerae* human infections (*Colwell et al., 2003*; *Huq et al., 2010*). Deletion of *vps* genes, which eliminates surface-biofilm formation in vitro, also reduced colonization in a mouse model of cholera disease (*Fong et al., 2010*). *V. cholerae* exists as planktonic cells and in debris-attached aggregates in stool samples obtained from human subjects (*Faruque et al., 2006*; *Nelson et al., 2007*). These observations suggest that the ability of *V. cholerae* to form multicellular communities (in liquid and on surfaces) is part of its infection and dispersal process. During infection, following colonization of the surface of the intestinal epithelium, *V. cholerae* grows to abundance, enters stationary phase, and then triggers a mucosal escape program, which depends on both the stationary-phase alternative sigma factor RpoS and the HCD QS-master regulator HapR (*Nielsen et al., 2006*). These steps enable re-entry of *V. cholerae* cells into the intestinal lumen. We propose that it is in this final stage of the infection cycle, after cells re-enter the intestinal lumen, that aggregation occurs because two conditions are met: the cells are in stationary phase and in the HCD QS-state. We speculate that formation of aggregates allows *V. cholerae* a superior mechanism (possibly by protecting aggregate-associated cells from chemical insults or by increasing intestinal transit rates) to survive passage through the small intestine and re-entry into the marine environment, where *V. cholerae* is immediately faced with a limited nutrient supply (*Kamp et al., 2013*). Moreover, aggregates may determine the rate at which *V. cholerae* passages through the intestine: work in larval zebrafish, a model amenable to live imaging, has shown that bacterial aggregates are commonly found in the intestinal lumen and increased aggregation is correlated with elevated rates of bacterial expulsion from the intestine (*Jemielita et al., 2014*; *Logan et al., 2018*; *Wiles et al., 2016*).

In support of this model, we note that half of the genes (*flgC*, *varS*, *sdhC*, *tolB*, *sspA*, and *pnp*) identified here as required for aggregation have previously been shown to be involved in *V. cholerae* colonization, mucosal penetration, or dissemination from the host (*Kamp et al., 2013*; *Liu et al., 2008*; *Merrell et al., 2002*). Live imaging in mice also shows that multicellular *V. cholerae* communities are present on epithelial surfaces and, moreover, these communities are clonal, while non-clonal aggregates are found in the intestinal lumen (*Millet et al., 2014*). This finding is consistent with the observation that VPS-dependent surface-biofilms are clonal (*Nadell et al., 2015*), while we show here that aggregates are non-clonal (*Figure 2—figure supplement 7*). Additionally, our screen identified that the *vc0175* gene, located within the VSP-I region, is required for aggregate formation. VSP-I and VSP-II are two genomic islands that distinguish the currently dominant biotype, El Tor, from the classical biotype (*Dziejman et al., 2002*). The El Tor biotype has supplanted the classical biotype as the primary cause of the pandemic disease cholera. Prior to the acquisition of these genomic islands, along with the El Tor CTX prophage and several additional point mutations, the El Tor *V. cholerae* biotype caused infections in humans, but lacked pandemic potential (*Hu et al., 2016*). Possibly, the ability to robustly form aggregates contributes to the current dominance of the El Tor

biotype by making either host infection or host dispersal more productive or by increasing environmental persistence. Future work in animal models is required to test these hypotheses.

Now we turn to *V. cholerae* aggregation in its other habitat: the marine environment. Aggregation could promote environmental persistence of *V. cholerae* by providing a mechanism for the rapid formation of multicellular communities under conditions of nutrient deprivation. Supporting this idea, our screen identified genes (*sspA, varS, sdhC, lexA, lrp, pnp*) with known roles in stress response or response to changes in carbon metabolism (*Brinkman et al., 2003*; *Butala et al., 2009*; *Lenz et al., 2005*; *Merrell et al., 2002*; *Romeo, 1998*; *Tsou et al., 2011*). With respect to the environment, aggregate formation might provide insight into conditionally viable environmental cells (CVEC; related to viable but not culturable cells (VBNC)) which are clumps of dormant environmental *V. cholerae* isolates that resist culturing except under very specific conditions (*Alam et al., 2007*; *Faruque et al., 2006*; *Kamruzzaman et al., 2010*). Non-clonal aggregate formation (*Figure 2—figure supplement 7*) may also provide *V. cholerae* a multicellular lifestyle amenable to horizontal gene transfer (HGT). The VPS-dependent surface-biofilm program likely cannot foster genetic diversity because, as mentioned above, *V. cholerae* surface-biofilms are clonal (*Nadell et al., 2015*). Expression of the genes encoding the competence machinery required for HGT is activated at HCD and in the presence of chitin (*Blokesch and Schoolnik, 2008*; *Meibom et al., 2005*). We suggest that aggregation could aid in the colonization of chitin surfaces and/or provide an alternative route to HGT. Finally, formation of *V. cholerae* aggregates during stationary phase has parallels to the spore-formation program in *Myxococcus xanthus* (*Shimkets, 1999*). Specifically, both processes occur under conditions of starvation and require population-level collective behavior to foster community-level benefits. Additionally, in the case of *V. cholerae*, aggregation might provide ecological advantages such as promoting environmental dissemination and concentrating biomass, for example, by altering the buoyancy of the community which could aid in movement through the water column. To reap these putative environmental benefits, *V. cholerae* must successfully confront associated challenges such as the emergence of cheaters that do not contribute to aggregate formation but nonetheless obtain the public good(s) the aggregate provides. We speculate that aggregation is driven by surface adhesins that additionally serve in self/non-self kin-recognition (*Smukalla et al., 2008*), which can defend communities against free-riders. Kin-recognition in *V. cholerae* can occur via the pilins TcpA and PilA (*Adams et al., 2018*; *Kirn et al., 2000*; *Taylor et al., 1987*). While the aggregation process that we report does not depend on either of these pili (*Figure 2—figure supplement 4*), *V. cholerae* may deploy additional, as-yet undiscovered, kin-recognition systems to control its community diversity in aggregates. Curiously, our screen identified genes required for aggregate formation, but it did not yield genes that encode obvious structural components required for *V. cholerae* aggregation. We are currently focused on identifying structural genes and on defining the mechanism underlying aggregate assembly. We anticipate that deeper understanding will provide insight into whether the aggregation program is specific to *V. cholerae* or is more broadly conserved among bacteria. Specifically, structural genes involved in *V. cholerae* aggregation could play analogous roles in other bacterial species. Additional studies in species known to aggregate should reveal if this is the case.

We have identified three relevant timescales for the aggregation program: the time at which *V. cholerae* commits to the aggregation program (7 h), the time by which aggregate formation occurs (by 22 h), and the timeframe over which aggregation occurs (<30 min). We discuss these three timescales in turn.

1. At 7 h, a time shortly before *V. cholerae* cultures enter stationary phase under our conditions, *V. cholerae* cells become refractory to the addition of autoinducers and commit to one of two developmental programs: the formation of aggregates, following a subsequent long delay, or to continuation as planktonic cells. We argue that this timepoint serves as a developmental checkpoint employed by *V. cholerae* to verify the execution of the optimal, cell-density-dependent, strategy for survival during stationary phase. For example, aggregation, a process whose kinetics must necessarily be driven by the encounter rate between bacteria in solution, will progress more efficiently when the cell density is high compared to when cell density is low.

2. By 22 h, *V. cholerae* undergoes the entire process of aggregation. Strikingly, the time by which aggregation occurs (22 h) is 15 h after the timepoint at which, from the context of QS signals, the cells have committed to this program. This quiescent period may provide *V. cholerae* a temporal window in which it can abort the aggregation program if, for example, new nutrient

sources become available. The genes that we identified in our screen were assayed for their contributions to aggregate formation at T = 22 h, and thus the possibility exists that they affect aggregation kinetics by delaying the onset of aggregate formation.

3. The process of aggregation occurs within 30 min. This timeframe is far more rapid than the formation of mature *V. cholerae* surface-biofilms, a process that can take up to 20 h under laboratory conditions (*Yan et al., 2016*). The rapidity of the aggregation process indicates that the underlying mechanism may be analogous to mechanisms driving aggregation in colloidal systems. For example, changes in the zeta potential on the cell surface may lead to aggregation (*Babick, 2016*). Alternatively, surface-exposed molecules may act as polymer brushes hindering aggregation until changes in the extracellular environment cause adjacent surfaces to entangle (*Chen et al., 2017*). By exploiting such physical processes, *V. cholerae* may be able to form aggregates in a rapid and metabolically efficient manner.

In conclusion, we have demonstrated a QS-controlled program of aggregation in *V. cholerae* that occurs in liquid and is independent of the surface-biofilm program. Further study of the formation of these multicellular communities may yield insight into the natural lifecycle of *V. cholerae*, cooperative strategies employed by *V. cholerae* to survive in its markedly different environmental niches, and broader mechanistic principles employed by bacteria that enable rapid multicellular community building to defend against predators or other harmful environmental factors, or that enable the collective to survive starvation.

## Materials and methods

### Bacterial strains and reagents

All *V. cholerae* strains used here were derived from a streptomycin-resistant variant of the WT O1 El Tor biotype C6706str2 (*Thelin and Taylor, 1996*). *Escherichia coli* strain S17-1λ*pir* was used for cloning. Antibiotics, when appropriate, were used at the following concentrations: ampicillin, 100 mg/L; kanamycin 100 mg/L; polymyxin B, 50 u/L; streptomycin, 500 mg/L. Tetracycline was used at 10 mg/L for strain construction and at 1 mg/L for plasmid maintenance. X-Gal was used at 50 mg/L. Chemical syntheses of CAI-1 and AI-2 have been previously described (*Higgins et al., 2007*; *Semmelhack et al., 2005*). Strains are listed in *Supplementary file 1*.

### DNA manipulation and strain construction

Standard molecular cloning techniques were used for plasmid construction (*Sambrook et al., 1989*). Primers are listed in *Supplementary file 2*. Chromosomal alterations in *V. cholerae* were performed using allelic exchange with pKAS32 (*Skorupski and Taylor, 1996*) or MuGENT (multiplex genome editing by natural transformation) (*Dalia et al., 2014a*). When using pKAS32, DNA fragments > 1 kB upstream and downstream of the genomic region to be deleted were amplified via PCR, fused using overlap extension PCR (OE-PCR) (*Ho et al., 1989*), and subsequently inserted into pKAS32 using ligation or Gibson assembly (*Gibson et al., 2009*). For MuGENT, approximately 3 kB regions upstream and downstream of the genomic region to be deleted were amplified via PCR. OE-PCR was subsequently used to fuse these fragments upstream and downstream of a DNA fragment encoding a Kan$^R$ cassette. The resulting product was provided to naturally competent *V. cholerae* cells grown on shrimp shells, as previously described (*Dalia et al., 2014b*; *Dalia et al., 2014a*). Following selection and isolation of colonies on LB plates containing both kanamycin and polymyxin B, the deletion was verified by PCR. For experiments in *Figure 4—figure supplement 2*, we used MuGENT for mutagenesis and co-transformed cells with a selectable marker at a neutral locus, *vc1807:Kan$^R$*, and a DNA fragment containing *lacZ:P$_{dns}$-dns*. Transformants were selected on LB plates containing kanamycin, polymyxin B, and X-Gal. *mKate2* (*Shcherbo et al., 2007*), *mTFP1* (*Ai et al., 2006*), or *mKO* (*Karasawa et al., 2004*) genes, each driven by *pTac*, were inserted onto the *V. cholerae* chromosome at the *lacZ* site, as previously described (*Nadell et al., 2015*).

### Aggregate formation

All growth media were filter sterilized (pore size: 0.22 μm). *V. cholerae* strains were grown overnight in LB (Fisher BioReagents, Pittsburgh, PA; Tryptone 10 g/L, yeast extract 5 g/L, and NaCl 10 g/L) at 37°C with shaking (250 rpm). Cultures were back-diluted 1:100 in LB supplemented with 10 mM

CaCl$_2$ (*Kierek and Watnick, 2003b*), and incubated at 37°C with shaking (250 rpm). After 1 h (approximate OD$_{600}$: 0.04), cultures were diluted 1:20 into 2 mL of LB +10 mM Ca$^{2+}$ in 20 mL Pyrex test tubes. Samples were incubated in the outer ring of a rolling drum (New Brunswick, Edison, NJ; model # M1053-4004; 1 Hz) at 30°C. At designated time points, 150 μL samples were removed from a fixed height within the culture and deposited into wells of a No. 1.5 coverslip 96-well microtiter dish (MatTek, Ashland, MA; part # P96G-1.5–5 F). The samples were dispensed into the microtiter wells using a single-channel electronic pipette (Eppendorf, Hamburg, Germany; Xplorer) set at the lowest possible aspiration and dispensation speed (172 μL/s). For aggregate formation in plastic-bottomed 24-well microtiter dishes, microplates were mounted on an orbital shaker (IKA, Staufen im Breisgau, Germany; KS 260 Basic; 250 rpm) and the samples were grown for 48 h at 30°C. The low surface-attachment assays used plastic-bottomed Corning Costar 24-well microtiter dishes (Corning, Corning, NY). Biological replicates are defined as aggregates derived from an isolated, individual colony. Technical replicates refer to samples taken from independent bacterial cultures.

## A genetic screen for factors promoting aggregation

Δ*vpsL* HCD-locked and Δ*vpsL lacZ:P$_{hapR}$-hapR* HCD-locked *V. cholerae* strains were mutagenized with Tn5 as previously described (*Miller et al., 2002*). Mutants were isolated on LB plates containing kanamycin and polymyxin B (<200 colonies per plate). The colonies were grown at 37°C for ~16 h to ensure that differences in colony opacity could be observed. Approximately 25 000 colonies were assessed in each of the two mutagenesis screens. Changes in colony opacity were determined by comparing the opacity of individual colonies to adjacent colonies on the same plate. All mutants exhibiting alterations in colony opacity were purified and isolated by restreaking on plates containing kanamycin and polymixin B. Aggregate formation was subsequently assayed at 22 h, using the protocol described above. No antibiotics were present in the growth medium for the aggregation assays. Transposon insertion sites were determined using arbitrary PCR (*Saavedra et al., 2017*). Images of colony opacity were obtained after 24 h of growth of colonies on LB plates at 37°C using a stereomicroscope (Leica, Wetzlar, Germany; M125; 20X zoom) equipped with a Leica MC170 HD camera and using a gooseneck light source to provide oblique sample illumination.

## Bioluminescence assay

*V. cholerae* cultures were grown using the above aggregate formation protocol and sampled at the indicated time points. Bioluminescence and OD$_{600}$ were respectively measured using a Tri-Carb 2810 TR scintillation counter (PerkinElmer, Waltham, MA) and a DU800 spectrophotometer (Beckman Coulter, Brea, CA). Prior to measuring OD$_{600}$, 1 mL of each culture was transferred to a 1.5 mL Eppendorf tube containing small acid-washed glass beads (Sigma-Aldrich, St. Louis, MO; model # G8772; 425–600 μm) and samples were vigorously shaken for 10 min on a vortex mixer to break apart aggregates.

## Microscopy and image analysis

Images were acquired with a Leica SP-8 point scanning confocal microscope equipped with a tunable white-light laser (Leica; model # WLL2; excitation window = 470–670 nm). mTFP1 was excited with a 442 nm continuous wave laser (Leica) and all other fluorophores were excited using the tunable white-light laser. Emitted light was detected using hybrid GaAsP spectral detectors (Leica, HyD SP) and timed gate detection was employed to minimize background signal. Aggregates were imaged using either a 10X air objective (Leica, HC PL FLUOTAR; NA: 0.30) or a 63X water immersion objective (Leica, HC PL APO CS2; NA: 1.20). All samples, unless specified otherwise, were imaged in the approximate center of the microtiter well.

Aggregate number and size were quantified using the 10X air objective with a field of view of 1163 × 1163 μm$^2$ (2048 × 2048 pixels$^2$). A total sample volume of 100 μm with a 2 μm step size was imaged, starting just above the coverslip surface. Resulting images were analyzed using custom software written in MATLAB, which is provided in a code repository (https://github.com/jemielita/aggregation.git; copy archived at https://github.com/elifesciences-publications/aggregation). In brief, to obtain segmented images using the 10X air objective, an intensity-based segmentation algorithm was applied to the 3D image stack followed by a minimum object size cutoff. To overcome sporadic under-segmentation of adjacent aggregates, the convexity of aggregates was exploited. For all

objects identified in a single plane, the root-mean-squared deviation (RMSD) between the area and convex area of the objects was computed. Subsequently, for all objects whose RMSD exceeded a cutoff, the shortest line across opposite quadrants of the object was computed and used to bisect the initially over-segmented object into two discrete objects. Following this procedure, 3D reconstructions of each aggregate were assembled by connecting overlapping 2D regions. As necessary, the results of this segmentation protocol were manually corrected. Within a given experiment, all parameters of the segmentation protocol were kept fixed. An exception was made for the analysis of the $\Delta vpsL \; \Delta xds \; \Delta dns$ HCD-locked strain for which the intensity threshold employed was lowered to properly segment the low-intensity distal (with respect to the objective) side of large aggregates. All quantitative imaging data reported in this manuscript were collected with the 10X air objective, with the exception of two datasets in *Figure 3—figure supplement 2*, which were collected with the 63X water objective. The 63X water objective was used to quantify cluster formation over a field of view of $1984 \times 1984 \; \mu m^2$ ($1984 \times 1984 \; pixels^2$). A total sample volume of 50 µm with a 1 µm step size was imaged, starting just above the coverslip surface. To segment images obtained with the 63X water immersion objective, analogous to what we describe above, an intensity-based segmentation algorithm was used, followed by the application of an upper cluster size cutoff.

For experiments in which the cross-sectional aggregate area was measured, subregions of the microtiter dish were imaged using the Leica Tile Scan module and an image of the full microtiter well was computationally assembled. As above, an intensity-based segmentation algorithm was applied to the resulting image, followed by a minimum object size cutoff.

We define the aggregate volume fraction as the sum of the volume of aggregates identified in the imaged volume normalized by the total volume imaged. We define the average aggregate volume as the effective average aggregate volume in which a bacterium is found:

$$\frac{\sum_{i=1}^{N} v_i \times v_i}{\sum_{i=1}^{N} v_i}$$

where $v$ is the volume of an individual aggregate and $N$ is the total number of aggregates identified within a sample. An identical approach was used to define the average cluster size obtained using the 63X water objective and for computing the average cross-sectional area. To compare the distribution of aggregate volumes or cross-sectional areas generated by different strains, we used a two-sample Kolmogorov–Smirnov test on the distribution of data pooled from all biological replicates. Aggregate volume fraction and average aggregate size are reported as the mean ± standard deviation (SD) for a minimum of three biological replicates.

## eDNA quantification and staining

To quantify bulk eDNA levels, 1 mL of cultures were transferred to 1.5 mL Eppendorf tubes containing small acid-washed glass beads (Sigma-Aldrich, catalog # G8772; 425–600 µm). Samples were vigorously shaken for 10 min on a vortex mixer followed by centrifugation for 1 min at 15 000 rpm. The clarified supernatants were filter sterilized (pore size: 0.22 µm), and DNA was extracted using the standard ethanol precipitation technique (*Ausubel et al., 2002*; *Seper et al., 2011*). Phase lock gel (Sigma-Aldrich, Dow Corning high-vacuum grease, catalog # Z273554) was used during phenol extraction. eDNA content was subsequently quantified using a NanoDrop One[C] (ThermoFisher, Waltham, MA; catalog # ND-ONE-W).

In DNase I supplementation experiments, cultures were prepared as described above and aliquoted into a glass-bottom 96-well microtiter dish (MatTek) to a final volume of 100 µL. The cultures were grown in the microtiter dish at 30°C on an orbital shaker (IKA, KS260; 350 rpm). DNase I (Sigma-Aldrich, catalog # D5025) was added to samples at T = 0 h at a concentration of 100 Kunitz units/mL (*Turnbull et al., 2016*). Staining of eDNA was accomplished using the nucleic acid stain TOTO-1 iodide (ThermoFisher, catalog # T3600; final concentration: 1 µM). Staining of the samples in *Figure 5D* and *Figure 3—figure supplement 4* was accomplished using SYTO-9 (ThermoFisher, catalog # S34854; final concentration: 2.2 µM). When using either stain, samples were deposited, as above, into wells of a No. 1.5 coverslip 96-well microtiter dish to which the appropriate stain was subsequently added and gently mixed.

## Acknowledgements

We thank all members of the Bassler group and Howard Stone for fruitful discussions and critical feedback. We particularly thank Jing Yan for providing plasmids and Amanda Hurley for assistance with plasmid and strain construction. This work was supported by the Howard Hughes Medical Institute, the Max Planck Society-Alexander von Humboldt Foundation, NSF Grant MCB-1713731, NIH Grant 2R37GM065859 (BLB), NIH Grant R01GM082938 (NSW), and NSF Grant PHY-1734030 (MJ).

## Additional information

### Funding

| Funder | Grant reference number | Author |
| --- | --- | --- |
| Howard Hughes Medical Institute | | Bonnie L Bassler |
| National Institute of General Medical Sciences | R01GM082938 | Ned S Wingreen |
| Alexander von Humboldt-Stiftung | | Bonnie L Bassler |
| National Science Foundation | MCB-1713731 | Bonnie L Bassler |
| National Institute of General Medical Sciences | 2R37GM065859 | Bonnie L Bassler |
| National Science Foundation | PHY-1734030 | Matthew Jemielita |

The funders had no role in study design, data collection and interpretation, or the decision to submit the work for publication.

### Author contributions

Matthew Jemielita, Conceptualization, Data curation, Software, Formal analysis, Validation, Investigation, Methodology, Writing—original draft, Project administration; Ned S Wingreen, Conceptualization, Supervision, Validation, Writing—review and editing; Bonnie L Bassler, Conceptualization, Resources, Data curation, Formal analysis, Supervision, Funding acquisition, Validation, Writing—original draft, Project administration

### Author ORCIDs

Matthew Jemielita http://orcid.org/0000-0002-9469-4087
Ned S Wingreen http://orcid.org/0000-0001-7384-2821
Bonnie L Bassler http://orcid.org/0000-0002-0043-746X

### Decision letter and Author response

Decision letter https://doi.org/10.7554/eLife.42057.043
Author response https://doi.org/10.7554/eLife.42057.044

## Additional files

### Supplementary files

• Supplementary file 1. Strain list
DOI: https://doi.org/10.7554/eLife.42057.039
• Supplementary file 2. Plasmid list
DOI: https://doi.org/10.7554/eLife.42057.040
• Transparent reporting form
DOI: https://doi.org/10.7554/eLife.42057.041

## Data availability

All data generated or analyzed during this study are included in the manuscript and supporting files. Source data files have been provided for all quantitative data.

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
