## [Decision Letter]

Thank you for submitting your article "Quorum sensing controls *Vibrio cholerae* multicellular aggregate formation" for consideration by *eLife*. Your article has been reviewed by three peer reviewers, and the evaluation has been overseen by Gisela Storz as the Senior and Reviewing Editor. The following individuals involved in review of your submission have agreed to reveal their identity: Kim Orth (Reviewer #1); Camille Danne (Reviewer #3).

The reviewers have discussed the reviews with one another and the Reviewing and Senior Editors have drafted this decision to help you prepare a revised submission.

Summary:

This manuscript examines programs of multicellularity in the bacterial pathogen *Vibrio cholerae* that are activated by quorum sensing. The authors study quorum sensing in two modes of multicellularity: non-surface-associated aggregates and surface-associated biofilms. The authors report that genes essential for surface-associated biofilm formation are not required for aggregate formation and that the quorum sensing master regulator HapR is essential for aggregation to occur. This is in contrast to surface-associated biofilm formation where HapR suppresses biofilm formation. The authors go on to show that extracellular DNA plays a role in aggregate formation. It is interesting that the authors observe that aggregates form from pre-existing cells whereas surface-associated biofilms are typically assembled through mechanisms that require clonal expansion. Finally, the authors performed a genetic screen to identify a small number of genes required for aggregation, although how these genes function to promote aggregation remains unknown. The authors discuss their observations in terms of the *V. cholerae* lifecycle and pathogenesis. They suggest that aggregated *V. cholerae* might be better able to survive re-entry into the environment and/or promote re-infection of a new host. Collectively, this work was well done and contributes to an alternative perspective on a long-standing question in the field: "What is a biofilm?"

Major comments:

Experimental:

1) None of the mutants tested have been complemented. The authors should at least show that the "aggregation" phenotype is restored once HapR is reintroduced in the *ΔhapR* strain, as the main message of the paper relies on this finding: aggregation is QS-dependent but relies on a different QS-circuit compared to surface-biofilm formation.

2) None of the graphs show statistical analysis. Even with low N numbers, the authors should use non-parametric tests to test the differences they observe.

3) Figure 4: The study would benefit from exploring the individual contributions of Xds and Dns DNases, particularly as only the latter has been shown to be regulated by HapR signaling, which this paper implicates in aggregate formation. To more clearly establish the contribution of eDNA to aggregate size and average aggregate volume, genetic complements of the deleted DNases, or biochemical complementation via application of exogenous DNase targeting eDNA, should also be conducted.

4) If feasible within a reasonable time frame (2-3 months), the authors should test their mutants in a zebrafish model to support their hypothesis about niche adaptation, for example comparing the colonization capacity of the wildtype strain versus the *ΔhapR* mutant strain. We realize that this experiment may be more involved, but experiments showing non-surface-associated aggregates affect pathogenesis would significantly increase the novelty of this study.

Editorial:

5) The authors should be better about putting their work in context of what has already been published.

- The work is reminiscent of existing studies – the authors do not really define a new program of multicellularity – the authors should soften such statements (as in the Abstract).

- Figure 2—figure supplement 6: Others have observed that aggregated bacteria are able to recruit single cells (Kragh et al., 2018), similar to what the authors describe.

- Figure 4: The authors show that eDNA is important for aggregation. This has been observed by others (for example, Schleheck et al., 2009).

- Figure 3: Another paper examined quorum-regulated self-aggregation in *B. thailandensis* upon exposure to nutrient-limiting conditions (Chandler et al., 2009). In that paper, quorum mutants did not aggregate, but aggregation could be restored upon addition of exogenous autoinducer.

6) Nomenclature: Engineers argue that aggregation is a general term that can and should be further defined (flocculation agglutination, coagulation, agglomeration…). Furthermore, the term aggregate is widespread in the biofilm literature (for example, see Thomas Bjarnsholt's work) and linking a new definition to the term aggregate will likely cause confusion in the field. Thus, it would be better if the authors did not associate an additional term with aggregation, especially when the underlying mechanism is unknown.

7) Discussion:

- The authors thoroughly described the relevance of the aggregation phenotype in *V. cholerae* and reference a similar phenomenon in *Myxococcus xanthus*. Because this suggests conservation of the aggregation phenotype across bacterial species, it would strengthen the paper to discuss in more detail whether and how aggregation and the proteins mediating it are conserved across Gram-negative bacteria.

- The authors illuminate multiple layers of temporal regulation underlying the aggregation phenotype: that QS autoinducers must be present by 7 hours for aggregation to occur, that aggregation occurs within a 30-minute timespan, and that aggregation begins after ~19.5-20 hours of growth. Because genes were assessed for their role in aggregate formation only at 22 hours, we suggest noting that the genes implicated in aggregate formation may also contribute to aggregation kinetics, as their deletion may have delayed aggregate formation instead of abrogating it entirely.

- A suggested addition to the outline for future work would be to explore possible redundancies in the molecular mechanisms underlying aggregation, which are to be expected particularly if aggregation is common to many Gram-negative bacteria and is important for the bacterial life cycle.

8) Other editorial suggestions:

- Briefly explain how "aggregate volume fraction" is measured/calculated.

- "VPS is dispensable for aggregate formation in liquid. In the remainder of the experiments reported here, unless explicitly stated otherwise, all strains harbor the Δ*vpsL* mutation." Explain why this was done.

- In the figure legends, it is mentioned "All strains harbor mKO constitutively expressed from the chromosome." Explain why this was done, particularly for readers who are not familiar with *V. cholerae*.

- Figure 2—figure supplement 2 legend: What are the genes *mKO, mKate2, mTFP*? What is the staining procedure used here? To what do the three colors correspond?

- The conclusion "aggregates are finite sized" is not supported by the Figure 2—figure supplement 2, or if it is the case, please clarify.

- "We deleted homologs of wbfF and wbfR, the genes implicated in the former study": Implicated in what? Not clear for readers who are not familiar with this former study.

- Subsection “Extracellular DNA contributes to aggregation”, second paragraph: Figure 4F shows a bigger difference between *ΔvpsL* HCD-locked and *ΔvpsL Δxds Δdns* HCD-locked than in Figure 4—figure supplement 1B. Why?

- Discussion, second paragraph: The authors talk about "aggregation" in the intestinal lumen. How can they be sure which in vitro phenotype it corresponds to? Could it not be "biofilm formation"?

9) Figure modifications:

- Figure 2: Include a series of images at 16, 19, 22, and 25 hours of true wild-type *V. cholerae* (with *vpsL*) to demonstrate that deletion of *vpsL* does not significantly alter aggregate formation. Though not integral to the message conveyed in Figure 2, inclusion of a supplemental figure of images at 16, 19, 22, and 25 hours of *vpsL*-complemented strains would elucidate whether deletion of *vpsL*, and not polar effects, caused the loss of small cluster formation in LCD-locked *V. cholerae*. This is especially pertinent when contrasting the mechanisms underlying LCD clustering and HCD aggregation.

- Figure 3: Add colored boxes (black and gray) to y-axis of graph in 3B, designating clearly what each colored bar is measuring.

---

## [Author Response]

Major comments:Experimental:1) None of the mutants tested have been complemented. The authors should at least show that the "aggregation" phenotype is restored once HapR is reintroduced in the ΔhapR strain, as the main message of the paper relies on this finding: aggregation is QS-dependent but relies on a different QS-circuit compared to surface-biofilm formation.

As suggested, in the revised work we have complemented the ∆*vpsL* ∆*hapR* HCD-lockedstrain by introducing a chromosomal copy of *hapR* at the *lacZ* locus. Figure 3—figure supplement 4 shows that the introduction of *hapR* restores the aggregation phenotype. The experiment and result are now described in the last paragraph of the subsection “Aggregation is HapR dependent”.

2) None of the graphs show statistical analysis. Even with low N numbers, the authors should use non-parametric tests to test the differences they observe.

We included summary statistics (mean and standard deviation) for all presented quantitative data in the original figure legends. Inspired by this feedback, we have now also included statistical analyses in the relevant places. In Figure 4G, in which we quantify eDNA content across strains, we apply a t-test to our data. In Figure 4F, Figure 4—figure supplement 2, and Figure 4—figure supplement 3, in which we compare the average aggregate sizes between ∆*vpsL* HCD-locked, and the ∆*vpsL* ∆*xds* ∆*dns* HCD-locked strains, we have now applied a Kolmogorov-Smirnov test to compare the size distributions in the different strains. Additionally, for these data sets, we realized that our initial approach to quantify aggregate volume within a given field of view was inadequate in specific cases. In particular, since the ∆*vpsL* ∆*xds* ∆*dns* HCD-locked strain produces aggregates of size comparable to that of the imaged region, our original approach could have provided a biased estimate of the average aggregate volume. To overcome this issue, in the revised manuscript, we have computed the average cross-sectional area for all aggregates within the microtiter well. This approach is laid out in the second paragraph of the subsection “Extracellular DNA contributes to aggregation”.

With due respect, we believe that it is not appropriate to employ the above statistical tests to compare our aggregating/non-aggregating conditions. Specifically, in cases in which the cells do not aggregate, we cannot measure the distribution of aggregate size, which would be a requirement for the application of statistical analysis to determine if other, aggregating samples differ from non-aggregating samples. However, to alleviate any concerns, in the revised manuscript, we now provide the data for individual aggregate volumes, rather than simply the summary statistics.

3) Figure 4: The study would benefit from exploring the individual contributions of Xds and Dns DNases, particularly as only the latter has been shown to be regulated by HapR signaling, which this paper implicates in aggregate formation. To more clearly establish the contribution of eDNA to aggregate size and average aggregate volume, genetic complements of the deleted DNases, or biochemical complementation via application of exogenous DNase targeting eDNA, should also be conducted.

The studies exploring the individual contributions of Xds and Dns were included in the submitted manuscript as Figure 4—figure supplement 1. These data show that the contribution of Dns to the increase in aggregate size is larger than that of Xds, and that the simultaneous deletion of both nucleases leads to the formation of even larger aggregation.

We thank the reviewers for suggesting these additional informative experiments that further support our conclusions. We performed both requested experiments and included the results in the revised manuscript.

Biochemical complementation with DNase I: Supplementation with DNase I at 100 Kunitz units per mL (a concentration previously used in the literature) reduced aggregation in the ∆*vpsL* ∆*xds* ∆*dns* HCD-locked strain to level of the ∆*vpsL* HCD-locked strain. Importantly, DNase I supplementation did not alter aggregation in the ∆*vpsL* HCD-locked strain. These data verify that eDNA contributes to, but is not essential for, aggregation. We discuss our new DNase I experiments in the subsection “Extracellular DNA contributes to aggregation", amended our Materials and methods subsection “Microscopy and image analysis”, and include a new figure as Figure 4—figure supplement 3.

Complementation of the ∆*vpsL* ∆*xds* ∆*dns* HCD-lockedstrain with exonuclease gene: Dns plays the major role in aggregation and Xds plays only a minor role (see Figure 4—figure supplement 1). For that reason, we focused our complementation efforts on Dns. We introduced a chromosomal copy of *dns* at the *lacZ* locus. Reintroduction of *dns* suppresses aggregation. New text explaining the experiment and the result is in the second paragraph of the subsection “Extracellular DNA contributes to aggregation” and the data are reported in new Figure 4—figure supplement 2.

Thus, both biochemical and genetic complementation restores the wild-type phenotype.

4) If feasible within a reasonable time frame (2-3 months), the authors should test their mutants in a zebrafish model to support their hypothesis about niche adaptation, for example comparing the colonization capacity of the wild-type strain versus the ΔhapR mutant strain. We realize that this experiment may be more involved, but experiments showing non-surface-associated aggregates affect pathogenesis would significantly increase the novelty of this study.

While we agree that these experiments would be valuable, we are unfortunately unable to carry them out. My lab has never worked with zebrafish so we have no expertise, we have no facilities or animals, nor do we have an existing collaboration in this area. Establishing such a model in larval zebrafish, could take years. We hope the Editor and reviewers can appreciate that such an effort is beyond the scope of this manuscript.

Editorial:5) The authors should be better about putting their work in context of what has already been published.- The work is reminiscent of existing studies – the authors do not really define a new program of multicellularity – the authors should soften such statements (as in the Abstract).- Figure 2—figure supplement 6: Others have observed that aggregated bacteria are able to recruit single cells (Kragh et al., 2018), similar to what the authors describe.- Figure 4: The authors show that eDNA is important for aggregation. This has been observed by others (for example, Schleheck et al., 2009).- Figure 3: Another paper examined quorum-regulated self-aggregation in B. thailandensis upon exposure to nutrient-limiting conditions (Chandler et al., 2009). In that paper, quorum mutants did not aggregate, but aggregation could be restored upon addition of exogenous autoinducer.

We thank the reviewers for bringing this literature to our attention. During the preparation of this manuscript we focused on placing our findings within the context of the *V. cholerae* literature, not within the full literature on bacterial aggregation. We now realize that this approach gave the impression that we were claiming to have discovered an entirely new bacterial multicellular program, rather than a program specific to *V. cholerae*. Therefore, in the revised work, we have modified our Abstract to make it clearer that we are describing only a program in *V. cholerae.* Likewise, in the Discussion, first paragraph, we include new text to place our findings in the context of studies of aggregation in other bacterial species.

6) Nomenclature: Engineers argue that aggregation is a general term that can and should be further defined (flocculation agglutination, coagulation, agglomeration…). Furthermore, the term aggregate is widespread in the biofilm literature (for example, see Thomas Bjarnsholt's work) and linking a new definition to the term aggregate will likely cause confusion in the field. Thus, it would be better if the authors did not associate an additional term with aggregation, especially when the underlying mechanism is unknown.

We respectfully disagree that the use of the term “aggregation” is not specific enough and would increase confusion in the literature. We specifically chose the term aggregation because it is used to describe different processes that lead to the adherence of cells to one another. We do not currently have another word that more accurately describes the process we are studying.

We do agree that the term we assign to the phenotype studied here should not conflict with the terms used for other multicellular programs in *V. cholerae*. To the best of our knowledge, the multicellular programs reported in *V. cholerae* are surface-biofilm formation, Ca^2+^-dependent biofilm formation, autoagglutination, and auto-aggregation.

In the last paragraph of the subsection “*V. cholerae* forms multicellular aggregates in the HCD QS-state”, we have carefully laid out why we are using the term “aggregation” to describe the phenotype we have discovered and we contrast it to surface-biofilm formation, Ca^2+^-dependent biofilm formation, autoagglutination, and autoaggregation.

7) Discussion:- The authors thoroughly described the relevance of the aggregation phenotype in *V. cholerae* and reference a similar phenomenon in *Myxococcus xanthus*. Because this suggests conservation of the aggregation phenotype across bacterial species, it would strengthen the paper to discuss in more detail whether and how aggregation and the proteins mediating it are conserved across Gram-negative bacteria.

Our intention in comparing the aggregation phenotype in *V. cholerae* and fruiting body formation in *Myxococcus xanthus* was to outline two strategies bacteria employ to collectively cope with conditions of starvation and other ecological challenges. We did not intend to claim that the *V. cholerae* aggregation program is conserved among Gram-negative bacteria. To clarify, we now lay out specific parallels between *V. cholerae* aggregation and the *M. xanthus* developmental program (Discussion, fifth paragraph). Also, we emphasize that because we have not yet identified the structural genes underlying aggregate formation, we do not yet know whether this new *V. cholerae* program is broadly conserved in bacteria.

- The authors illuminate multiple layers of temporal regulation underlying the aggregation phenotype: that QS autoinducers must be present by 7 hours for aggregation to occur, that aggregation occurs within a 30-minute timespan, and that aggregation begins after ~19.5-20 hours of growth. Because genes were assessed for their role in aggregate formation only at 22 hours, we suggest noting that the genes implicated in aggregate formation may also contribute to aggregation kinetics, as their deletion may have delayed aggregate formation instead of abrogating it entirely.

We thank the reviewers for this suggestion. We have modified the Discussion accordingly.

- A suggested addition to the outline for future work would be to explore possible redundancies in the molecular mechanisms underlying aggregation, which are to be expected particularly if aggregation is common to many Gram-negative bacteria and is important for the bacterial life cycle.

We have added text to the fifth paragraph of the Discussion, to bring up the possibility that genes involved in aggregation in *V. cholerae* could have analogous roles in other bacterial species and that additional studies in species known to aggregate could determine whether this is the case.

8) Other editorial suggestions:- Briefly explain how "aggregate volume fraction" is measured/calculated.

A more specific definition of aggregate volume fraction is now provided in the manuscript on the last paragraph of the subsection “Microscopy and image analysis”. To further aid readers in understanding this definition, in the revised text we have also provided a brief definition at the first mention of aggregate volume fraction, in Figure 2. In the Figure 2 legend, we also now direct the reader to the Materials and methods section for the full description of how our analyses were performed.

- "VPS is dispensable for aggregate formation in liquid. In the remainder of the experiments reported here, unless explicitly stated otherwise, all strains harbor the ΔvpsL mutation." Explain why this was done.

We show early in the manuscript that VPS is not required for the aggregation process we study. Therefore, using strains that are Δ*vpsL* allows us to distinguish aggregation from the surface-biofilm program, for which VPS is essential. We have amended the text in the last paragraph of the subsection “*V. cholerae* forms multicellular aggregates in the HCD QS-state”, to better lay out this logic.

- In the figure legends, it is mentioned "All strains harbor mKO constitutively expressed from the chromosome." Explain why this was done, particularly for readers who are not familiar with *V. cholerae*.

We agree that the strategy we used for imaging should have been made clearer in the text. mKO is a fluorescent protein. We engineered our *V. cholerae* strains to constitutively express this protein to enable confocal imaging. We have modified the sentence in the Figure 3 legend to make this strategy clear. Additionally, in the first paragraph of the subsection “*V. cholerae* forms multicellular aggregates in the HCD QS-state”, we now state that we are using constitutive fluorescent proteins to visualize *V. cholerae* cells.

- Figure 2—figure supplement 2 legend: What are the genes mKO, mKate2, mTFP? What is the staining procedure used here? To what do the three colors correspond?

mKO, mKate2, mTFP are fluorescent proteins that, respectively, emit yellow, red, and blue light. No staining is used in this figure. We now specifically state this in the Figure 2—figure supplement 2 legend. Additionally, in the two locations in the manuscript where staining was exclusively used, Figure 5 and Figure 4—figure supplement 2, this approach has been noted.

- The conclusion "aggregates are finite sized" is not supported by the Figure 2—figure supplement 2, or if it is the case, please clarify.

We show in Figure 2—figure supplement 2 the results of experiments in which we independently grew cells harboring different fluorescent proteins, then, *after* aggregate formation, we mixed the samples together. Images of these mixed cultures show aggregates of only one color. The size (and morphology) of each aggregate is comparable, and moreover, comparable to those formed from a strain expressing only a single fluorescent protein. If aggregates had exhibited large variations in size we would have observed different-sized patches of single colors in the experiment shown in Figure 2 rather than discrete, single-color regions. We have clarified this observation in the figure legend to Figure 2—figure supplement 2. Also, we now include further support for this argument by providing, as Video 1 a *z*-scan through a culture containing aggregates, which demonstrates the extent of individual aggregates.

- "We deleted homologs of wbfF and wbfR, the genes implicated in the former study": Implicated in what? Not clear for readers who are not familiar with this former study.

In the last paragraph of the subsection “HCD-QS aggregates are VpsL-independent”, we have clarified the discussion of this previous study and the roles of the *wbfF* and *wbfR* genes.

- Subsection “Extracellular DNA contributes to aggregation”, second paragraph: Figure 4F shows a bigger difference between ΔvpsL HCD-locked and ΔvpsL Δxds Δdns HCD-locked than in Figure 4—figure supplement 1B. Why?

The experiments in Figure 4F and Figure 4—figure supplement 1B are independent experiments performed on different days. We attribute the difference to variation in the sample aggregate volume fraction between biological replicates.

- Discussion, second paragraph: The authors talk about "aggregation" in the intestinal lumen. How can they be sure which in vitro phenotype it corresponds to? Could it not be "biofilm formation"?

In the *V. cholerae* literature, biofilm formation is used to describe bacterial communities bound to surfaces, such as the intestinal epithelium. We are proposing a scenario whereby non-surface-associated aggregation contributes to the *V. cholerae* infection cycle, particularly, dispersal from the host back into the environment. Our reasoning relies on our demonstration that *V. cholerae* forms aggregates in the absence of surfaces coupled with our identification of genes required for this aggregation process having previously documented essential roles in intestinal colonization/dispersal.

9) Figure modifications:- Figure 2: Include a series of images at 16, 19, 22, and 25 hours of true wild-type *V. cholerae* (with vpsL) to demonstrate that deletion of vpsL does not significantly alter aggregate formation. Though not integral to the message conveyed in Figure 2, inclusion of a supplemental figure of images at 16, 19, 22, and 25 hours of vpsL-complemented strains would elucidate whether deletion of vpsL, and not polar effects, caused the loss of small cluster formation in LCD-locked *V. cholerae*. This is especially pertinent when contrasting the mechanisms underlying LCD clustering and HCD aggregation.

We thank the reviewers for suggesting this important control. A series of images of aggregation over time in wild-type *V. cholerae* has now been included as Figure 2—figure supplement 5. Wild-type *V. cholerae* exhibits similar patterns of aggregation as a ∆*vpsL* strain. Text has been updated in the first paragraph of the subsection “Aggregation dynamics are rapid”.

*vpsL* is one of 18 genes, located in two operons that are required for production of vibrio polysaccharide (VPS). The Δ*vpsL* mutation in our strain is certainly polar on the 6 genes that lie downstream of it in operon II. Thus, one cannot simply add back *vpsL* and restore VPS production. The biosynthetic pathway that produces VPS has never been determined, and to the best of our knowledge no one has successfully constructed a strain with *vpsL* in trans that restores biofilm formation. Presumably, this is the result of issues arising from complex regulation of VPS production, feedback loops, and difficulty in ensuring the correctly-balanced production of biosynthetic components, which preclude simple complementation studies.

- Figure 3: Add colored boxes (black and gray) to y-axis of graph in 3B, designating clearly what each colored bar is measuring.

A colored box has been added to Figure 3B as suggested.